# MicroRNAs mediate precise control of spinal interneuron populations to exert delicate sensory-to-motor outputs

**Shih-Hsin Chang[1,2,3], Yi-Ching Su[2], Mien Chang[2], Jun-An Chen[2,3]***

[1]Taiwan International Graduate Program in Interdisciplinary Neuroscience, National Yang-Ming University and Academia Sinica, Taipei, Taiwan; [2]Institute of Molecular Biology, Academia Sinica, Taipei, Taiwan; [3]Neuroscience Program of Academia Sinica, Academia Sinica, Taipei, Taiwan

**Abstract** Although the function of microRNAs (miRNAs) during embryonic development has been intensively studied in recent years, their postnatal physiological functions remain largely unexplored due to inherent difficulties with the presence of redundant paralogs of the same seed. Thus, it is particularly challenging to uncover miRNA functions at neural circuit level since animal behaviors would need to be assessed upon complete loss of miRNA family functions. Here, we focused on the neural functions of MiR34/449 that manifests a dynamic expression pattern in the spinal cord from embryonic to postnatal stages. Our behavioral assays reveal that the loss of MiR34/449 miRNAs perturb thermally induced pain response thresholds and compromised delicate motor output in mice. Mechanistically, MiR34/449 directly target *Satb1* and *Satb2* to fine-tune the precise number of a sub-population of motor synergy encoder (MSE) neurons. Thus, MiR34/449 fine-tunes optimal development of Satb1/2[on] interneurons in the spinal cord, thereby refining explicit sensory-to-motor circuit outputs.

*For correspondence:
jachen@imb.sinica.edu.tw

**Competing interests:** The authors declare that no competing interests exist.

## Introduction

MicroRNAs (miRNAs) are a class of small regulatory non-coding RNAs that participate in various biological functions by repressing gene expression post-transcriptionally (*Ambros, 2004*; *Bartel, 2004*; *He and Hannon, 2004*; *Kim, 2005*). Several mouse knockout (KO) studies of *Dicer*, the pivotal RNase for generating mature miRNAs, have revealed that global depletion of these small regulatory RNAs leads to obvious defects in embryonic development, from cell differentiation to animal survival, underlining the functional importance of miRNAs during embryonic development (*Bernstein et al., 2003*; *Chen et al., 2011*; *Harfe et al., 2005*; *Kanellopoulou et al., 2005*; *Tung et al., 2015*). However, loss-of-function studies on most individual miRNAs have revealed minor or no overt developmental or behavioral phenotypes in multiple organisms, partly reflecting the functional redundancy of miRNAs due to their polycistronic and paralogous features (*Alvarez-Saavedra and Horvitz, 2010*; *Miska et al., 2007*; *Olive et al., 2015*; *Park et al., 2012*). Moreover, a single miRNA can target multiple mRNAs, and a targeted mRNA can also be bound by different miRNAs at its 3' untranslated region (3' UTR) with differential stoichiometric affinities, representing a reciprocal one-to-multiple regulatory mechanism (*Bartel, 2009*; *Krek et al., 2005*; *Loh et al., 2011*). These multiple interactions with differential affinities and stoichiometries add complexity to functional redundancy, rendering it exceedingly difficult to study the function of individual miRNAs. Hence, considering the extensiveness of miRNA regulatory networks as well as their buffering effects, unequivocal functional assessments of a miRNA can only be established upon complete removal of all associated redundant miRNAs and pathways. Fortunately, the advent of CRISPR-Cas9-

**eLife digest** The spinal cord is an information superhighway that connects the body with the brain. There, circuits of neurons process information from the brain before sending commands to muscles to generate movement. Each spinal cord circuit contains many types of neurons, whose identity is defined by the set of genes that are active or 'expressed' in each cell.

When a gene is turned on, its DNA sequence is copied to produce a messenger RNA (mRNA), a type of molecule that the cell then uses as a template to produce a protein. MicroRNAs (or miRNAs), on the other hand, are tiny RNA molecules that help to regulate gene expression by binding to and 'deactivating' specific mRNAs, stopping them from being used to make proteins.

Mammalian cells contain thousands of types of microRNAs, many of which have unknown roles: this includes MiR34/449, a group of six microRNAs found mainly within the nervous system. By using genetic technology to delete this family from the mouse genome, Chang et al. now show that MiR34/449 has a key role in regulating spinal cord circuits.

The first clue came from discovering that mice without the MiR34/449 family had unusual posture and a tendency to walk on tiptoe. The animals were also more sensitive to heat, flicking their tails away from a heat source more readily than control mice.

At a finer level, the spinal cords of the mutants contained greater numbers of cells in which two genes, Satb1 and Satb2, were turned on. Compared to their counterparts in control mice, the Satb1/2-positive neurons also showed differences in the rest of the genes they expressed. In essence, these neurons had a different genetic profile in MiR34/449 mutant mice, therefore disrupting the neural circuit they belong to.

Based on these findings, Chang et al. propose that in wild-type mice, the MiR34/449 family fine-tunes the expression of Satb1/2 in the spinal cord during development. In doing so, it regulates the formation of the spinal cord circuits that help to control movement. More generally, these results provide clues about how miRNAs help to determine cell identities; further studies could then examine whether other miRNAs contribute to the development and maintenance of neuronal circuits.

mediated technology has made it more feasible to study miRNA function in mouse models (*Li et al., 2017*).

Many miRNAs are clustered with redundant paralogs in the genome. One well-characterized example of a polycistronic miRNA family is MiR34/449, which comprises six homologous miRNAs (MiR34a, 34b, 34c, 449a, 449b, and 449c) located at three different genomic loci—*Mir34a*, *Mir34b/c*, and *Mir449a/b/c* (*Mir449*)—that encode evolutionary-conserved sequences among vertebrates (*Olive et al., 2015*; *Song et al., 2014*). In this study, we use '*Mir34*' and 'MiR34' to indicate the mouse gene and the mature form of miRNA, respectively (*Desvignes et al., 2015*). The MiR34a, MiR34b, and MiR34c (MiR34) miRNAs appear to be direct targets of p53, and they have been shown to be involved in p53-mediated apoptosis in cancer contexts (*Bommer et al., 2007*; *Corney et al., 2007*; *He et al., 2007*; *Raver-Shapira et al., 2007*). Whereas *Mir34a* and *Mir34bc* are not located within other genes, the *Mir449* cluster is embedded in the second intron of a host gene, *Cdc20b* (*cell division cycle 20 homologue B*), and similar expression patterns of MiR449 and Cdc20b in mouse testis support that they may be subjected to the same mechanism of transcriptional regulation (*Bao et al., 2012*). Previous studies have demonstrated that the p53-mediated MiR34 and E2F1-mediated MiR449 pathways both target pro-survival genes to regulate cell fate determination (*Lizé et al., 2011*; *Lizé et al., 2010*). Apart from this functional redundancy in the context of DNA damage, an additional role for MiR449 in regulating multiciliogenesis by suppressing the Notch pathway has been demonstrated (*Lizé et al., 2011*; *Marcet et al., 2011*). Mature MiR34a has been shown to be ubiquitously expressed in a variety of organs, whereas MiR34bc and MiR449 exhibit a cell-type-specific expression pattern, predominantly in ciliated cells, such as in respiratory and reproductive tissues, as well as in the ependymal cells enriched with cilia responsible for cerebrospinal fluid flow within the CNS (*Bao et al., 2012*; *Kasai et al., 2016*; *Olive et al., 2015*). These reports highlight that MiR34/449 might have a functional significance in ciliogenesis (*Chevalier et al., 2015*; *Concepcion et al., 2012*; *Otto et al., 2017*; *Song et al., 2014*; *Wu et al., 2014*). Due to their shared

and redundant seed sequences, MiR34/449 displays compensatory expression upon single KO in a context-dependent manner, emphasizing the necessity to enact combinatorial KO of all *Mir34/449* alleles (*Bao et al., 2012*; *Choi et al., 2017*; *Concepcion et al., 2012*; *Wu et al., 2014*).

Based on three studies of *Mir34bc* and *Mir449* double KO (DKO) mice, as well as *Mir34/449* triple KO (TKO) mice, the most prominent phenotype arising from MiR34/449 ablation is the impairment of motile ciliogenesis, which results in respiratory dysfunction, male/female infertility, and postnatal lethality (*Otto et al., 2017*; *Song et al., 2014*; *Wu et al., 2014*; *Yuan et al., 2019*). Given that the MiR34/449 family is strongly expressed in the CNS, it is rather surprising that their function there has been largely unexplored, with only a potential role for MiR34/449 in regulating cortex development having been revealed in embryos (*Fededa et al., 2016*). In fact, information on the postnatal and adult functions of MiR34/449 has been lacking, perhaps because *Mir34/449* TKO mice exhibit a low survival rate, so acquiring sufficient numbers of adult *Mir34/449* TKO mice for behavioral assay is arduous. We previously uncovered that MiR34/449 exhibits a dynamic expression pattern during spinal cord development (*Li et al., 2017*), so we hypothesized that MiR34/449 might exert an as yet uncharacterized function in the spinal cord. In this study, we tested that hypothesis for embryonic, postnatal and adult mice.

The spinal cord is one of the most important executive centers for body movement, conveying somatosensory information and integrating motor commands emanating from spinal cord *per se* or the brain (*Arber, 2017*; *Arber and Costa, 2018*). The sophisticated processing and integration of sensory and motor circuits relies on proper organization of multiple neuron types in the spinal cord that possess related anatomical functions (*Osseward and Pfaff, 2019*). Interneurons (INs) that reside in the dorsal horn of the spinal cord terminate in exteroceptive sensory fibers and mediate sensory processing. The ventral horn contains both motor neurons (MNs) and various INs, which are critical for motor function and for generating the rhythmicity of motor behaviors (*Dasen, 2018*; *Garcia-Campmany et al., 2010*). The hallmarks of neuron populations in the intermediate region of the dorso-ventral spinal cord have just emerged following identification of specific markers (*Dobrott et al., 2019*). This region serves as a relay station that receives multiple convergent inputs from proprioceptive, exteroceptive, and corticospinal neurons, which connect monosynaptically to MNs (*Levine et al., 2014*; *Tripodi et al., 2011*). Activation of these premotor neurons, denoted motor synergy encoders (MSE), induces multiple motor pools that contribute to complex movements (*Levine et al., 2014*; *Osseward and Pfaff, 2019*). These sensory feedback pathways are indispensable for proper formation of spinal circuits and for motor execution, and their perturbation can result in defects of limb position, or imprecise or stiff movement (*Gatto et al., 2019*; *Koch et al., 2018*; *Osseward and Pfaff, 2019*). Among MSE, Satb1/2$^{on}$ neurons, a sub-population of lamina V/VI and III sensory relay neurons, represent a family of typical intermediate INs that control proper circuit development in a core sensory-to-motor spinal network. Conditional ablation of *Satb2* from the intermediate spinal cord results in a hyperflexion phenotype upon triggering the limb withdrawal reflex (*Hilde et al., 2016*). Although expression of *Satb2* is quite dynamic during spinal development, how generation of Satb2$^{on}$ INs controls precise sensory-to-motor output during development remains unknown.

It is well documented that miRNAs are critical regulators during embryonic spinal cord development, playing roles in progenitor patterning, subtype specification, functional maintenance, and neural regeneration (*Chen et al., 2011*; *Chen and Wichterle, 2012*; *Chen and Chen, 2019*; *Kye and Gonçalves, 2014*; *Tung et al., 2015*; *Wu et al., 2012*; *Zhu et al., 2013*). Furthermore, emerging evidence also demonstrates that miRNAs are not only required for early development but that they also display critical postnatal or adult-specific functions (*Sun and Lai, 2013*). Notably, loss of *Dicer* from late-stage post-mitotic spinal MNs using ChAT-Cre results in development of a spinal muscular atrophy (SMA)-like phenotype in a mouse model (*Haramati et al., 2010*). SMA is an autosomal recessive MN disease causing early childhood death. Furthermore, mice in which *Dicer* has been knocked out from proprioceptive sensory neurons display impaired maintenance of monosynaptic sensory-motor circuits in the spinal cord (*Imai et al., 2016*). However, it remains unclear if any miRNA participates in IN development and maintenance, or if miRNAs participate at neural circuit and physiological levels. Given that MiR34/449 expression varies dynamically during spinal cord development, we were prompted to test if this miRNA family is important for MN and IN development, as well as in the operation of sensory-motor circuits in the spinal cord.

To address these possibilities, we generated *Mir34bc/449* DKO and *Mir34/449* TKO mouse models, and then systematically investigated their phenotypes relating to CNS development. First, we uncovered low survival of postnatal and adult *Mir34bc/449* DKO and *Mir34/449* TKO mice. However, we applied several behavioral tests to assay spinal functions in survivors and observed several unusual phenotypes, including blepharoptosis, abnormal posture with back-arching, hind tiptoe walking, jumpy movement, and stimulant-induced hypersensitivity. Although motor systems of *Mir34bc/449* DKO or *Mir34/449* TKO mice only mildly compromised, we found that the MiR34/449 mutant mice displayed a more hypersensitive threshold in response to thermally induced pain stimulation. Mechanistically, we observed that the number of Satb1/2-expressing INs increased in the spinal cords of *Mir34/449* KO mice. This Satb1/2 increment is potentially targeted by MiR34/449 via direct binding in the spinal cord. Our findings demonstrate that MiR34/449 depletion results in a change to a spinal IN population that might contribute to altered sensory-motor responses, thereby revealing an important role for MiR34/449 in regulating the sensory-motor circuitry.

## Results

### MiR34/449 members are highly expressed in the developing and adult spinal cord

In previous studies, we identified a series of specific developmental stage enriched miRNAs during spinal cord development by taking advantage of an embryonic stem cell (ESC) differentiation approach (*Chen et al., 2011*; *Wichterle et al., 2002*). The MiR34/449 family particularly drew our attention as NanoString miRNA profiling revealed a very dynamic and distinct expression pattern, even they have the same seed sequence (*Li et al., 2017*; *Figure 1A*). Despite other studies having illustrated the expression patterns and functions of MiR34/449 in lung tissues and germ cells (*Concepcion et al., 2012*; *Otto et al., 2017*; *Song et al., 2014*; *Wu et al., 2014*), it remained enigmatic if these miRNAs are expressed in embryonic and adult neural tissues and what were their potential functions in the CNS. To address this knowledge gap, we systematically compared expression of MiR34/449 among the CNS (neocortex, cerebellum, and spinal cord), peripheral neural tissues (trigeminal ganglion and dorsal root ganglion), and non-neural tissues (heart, liver, and muscle) at an adult stage (postnatal day P50; *Figure 1B*~1F) by quantitative RT-PCR (RT-qPCR). We found MiR449b-5p to be undetectable in most tissues. Generally, the guide strands of the remaining five MiR34/449 members were enriched in neural tissues relative to non-neural tissues (*Figure 1B*~1F). Although MiR34a-5p is expressed in a variety of organs, it was much more abundant in neural than non-neural tissues. Notably, MiR34a-5p exhibited the highest expression level in the spinal cord (*Figure 1B*). This greater neural tissue enrichment was also apparent for MiR34b-5p, MiR34c-5p, MiR449a-5p, and MiR449c-5p (*Figure 1C*~1F).

This pattern of neural tissue enrichment prompted us to scrutinize expression profiles of MiR34/449 for embryonic (embryonic day E11.5~E17.5) to postnatal (P1~P28) stages by RT-qPCR (*Figure 1G*~1K). Generally, MiR34a-5p expression increased gradually across postnatal stages, with MiR34b-5p and MiR34c-5p also displaying a propensity for postnatal enrichment (*Figure 1G*~1I). Conversely, MiR449a-5p and MiR449c-5p were enriched at embryonic stages, manifesting much lower expression levels at postnatal stages when compared to MiR34a/b/c (*Figure 1J and K*).

To further determine the spatial distribution of MiR34/449 expression in mouse spinal cord, we assessed *in situ* hybridization of MiR34a-5p, representing the most abundantly expressed member of the MiR34/449 family in postnatal spinal cords (*Figure 1L and M*). To label neurons in the spinal cord or adjacent regions, we performed immunostainings using pan-neuronal (NeuN) and mature MN (ChAT) markers after *in situ* hybridization. For both embryonic and postnatal stages, we found that MiR34a-5p is expressed in most spinal cord neurons (*Figure 1M*), including ventral MNs (*Figure 1N*) and most dorsal INs (*Figure 1O*). These RT-qPCR and *in situ* hybridization results demonstrate that MiR34/449 members are strongly expressed in the CNS, most prominently in the spinal cord, albeit with a very dynamic expression pattern during spinal cord development. Among the MiR34/449 members, MiR34a-5p exhibits the strongest expression in the spinal cord, with low embryonic-stage expression and increasing levels as postnatal stages progress.

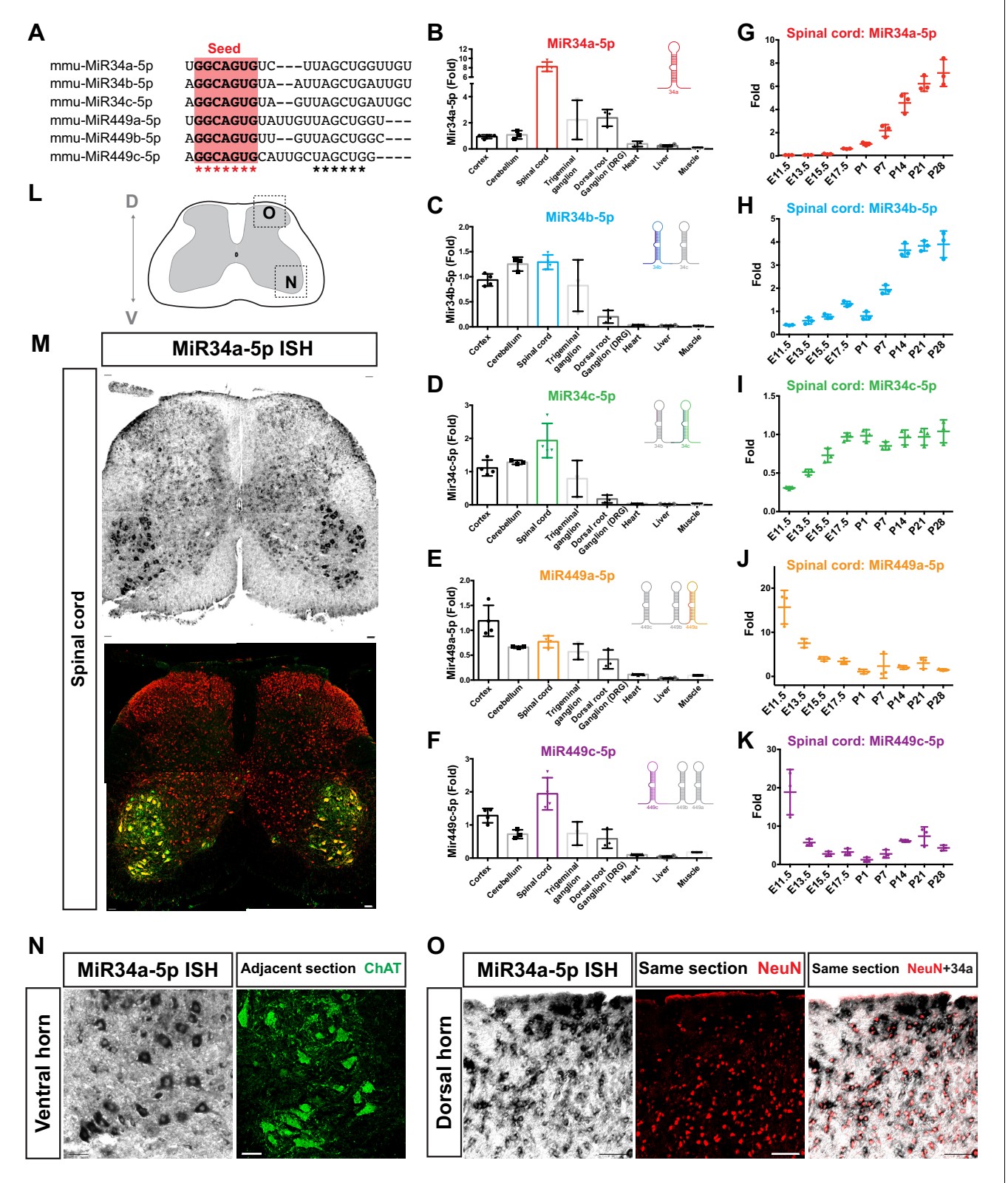

**Figure 1.** MiR34/449 miRNAs are expressed in the CNS and show dynamic expression profiles in the developing spinal cord. (A) Sequence alignment of mouse MiR34/449 miRNAs with asterisks indicating identical or conserved nucleotides. The seed sequences are highlighted in red. (B–F) Expression levels of MiR34/449 (MiR34a-5p, MiR34b-5p, MiR34c-5p, MiR449a-5p, and MiR449c-5p), as measured by RT-qPCR in multiple tissues of adult wild-type (WT) mice at postnatal day P50. The MiR34/449 miRNAs were first normalized against ubiquitously expressed MiR16 in the corresponding samples

*Figure 1 continued on next page*

Figure 1 continued

(delta Ct). The relative level of each miRNA was then normalized with that of cortex (delta-delta Ct) and is represented as fold-change (data represent mean ± SD from N ≥ 3 independent biological samples). (G–K) Expression levels of MiR34/449 miRNAs, as determined by RT-qPCR at various developmental time-points in the spinal cord. The relative level of each miRNA was first normalized against the ubiquitously expressed MiR16 (delta Ct), and then further normalized against that at the P1 time-point (delta-delta Ct) and is represented as fold-change (data represent mean ± SD from N = 3 independent biological samples). Note that MiR34/449 miRNAs show dynamic expression profiles during spinal cord development, with both MiR34a-5p and MiR34b/c-5p showing increasing levels, whereas MiR449a-5p and MiR449c-5p show decreasing levels. (L) Schematic illustration of a transverse spinal section across the dorso-ventral axis. (M) In situ hybridization (ISH) of MiR34a-5p with immunostaining for NeuN/ChAT in adjacent sections of the lumbar spinal cord at P20. (N and O) High-magnification images of the ventral (N) and dorsal (O) spinal cord, showing expression of MiR34a-5p in ventral MNs and dorsal INs. Scale bars represent 100 μm in (M) and 50 μm in (N) and (O).

The online version of this article includes the following figure supplement(s) for figure 1:

**Figure supplement 1.** Generation and genotyping analyses of *Mir449* KO mice by a CRISPR-Cas9 approach.

## *Mir34bc/449* DKO and *Mir34/449* TKO mice exhibit profound neurological disorders

Motivated by our above-described findings, we hypothesized that the MiR34/449 miRNAs may play critical roles in the developing spinal cord. As MiR34/449 members display extensive homology and a given miRNA family typically manifests dominant cell-type-specific expression (*Olive et al., 2015*), we generated a *Mir34/449* TKO mouse line in which all six family members were completely deleted from the *Mir34a*, *Mir34bc*, and *Mir449abc* loci (*Figure 2A*). To do that, previously generated *Mir34a*-floxed and *Mir34bc*-/- mice (*Concepcion et al., 2012*) were first interbred with germ line Cre mice (E2a-Cre) to acquire *Mir34a*-/-;*Mir34bc*-/- DKO mice. Consistent with previous studies (*Bao et al., 2012*; *Concepcion et al., 2012*), we did not observe an overt phenotype in this DKO mouse line (data not shown).

Subsequently, we generated *Mir449*-/- mice via a CRISPR-Cas9 approach with the intention of interbreeding it with our *Mir34a*-/-;*Mir34bc*-/- DKO mice to acquire the TKO (*Mir34a*-/-;*Mir34bc*-/-; *Mir449*-/-) line (*Figure 1—figure supplement 1A*) (see Materials and methods for details). Given that the gene encoding the *Mir449* cluster is embedded in the second intron of a host gene, *Cdc20b*, we designed two single-guide RNAs (sgRNAs) flanking the *Mir449* cluster in exon 2 and exon 3 of *Cdc20b* (*Figure 1—figure supplement 1A*), and then confirmed complete deletion of *Mir449* by sequencing and genotyping (*Figure 1—figure supplement 1B and C*). RT-qPCR further confirmed that expression of MiR449 had been abolished (*Figure 1—figure supplement 1D*). Although we recorded elevated *Cdc20b* mRNA levels specifically in testis and lung of our MiR449-deficient mouse model (*Figure 1—figure supplement 1E*), we did not observe overt defects in survival or fertility for either male or female mice. Moreover, we noticed compensatory expression of MiR34bc and MiR449 expression in the spinal cords of *Mir34a*-/-;*Mir34bc*-/- and *Mir34a*-/-;*Mir449*-/- compound KO mice (*Figure 1—figure supplement 1F*~H), supporting the desideratum to generate complete MiR34/449 KO mice to assess the functions of that miRNA family in the spinal cord.

To achieve this, mice harboring one wild-type (WT) allele of the *Mir34/449* family were then intercrossed to generate *Mir34/449* TKO or *Mir34bc/449* DKO mice (*Figure 2A*). Consistent with previously reported phenotypes for these lines, both *Mir34/449* TKO and *Mir34bc/449* DKO mice exhibited partial postnatal lethality and growth retardation with reduced body weight (*Figure 2B and C*; *Fededa et al., 2016*; *Song et al., 2014*; *Wu et al., 2014*). Therefore, we used either *Mir34/449* TKO or *Mir34bc/449* DKO mice for further analyses and other genetic combinations were employed as littermate controls (Ctrl) (*Figure 2A* and *Supplementary file 1f*).

Unexpectedly, both the *Mir34bc/449* DKO and *Mir34/449* TKO mice presented two waves of postnatal mortality: a first wave occurred after the first week (P7~P14) (*Song et al., 2014*), with a second wave between P21 and P28 (*Figure 2B*). Approximately 50% of *Mir34bc/449* DKO and *Mir34/449* TKO mice died within the first 2 weeks of birth, with an additional ~40% died within 4 weeks (*Figure 2B*). Consequently, only ~10% of mice survived to adulthood with less obvious growth retardation or abnormalities. This pattern of mortality prompted us to further investigate several of the late postnatal (P14~P28) phenotypes. For instance, we found that *Mir34bc/449* DKO and *Mir34/449* TKO mice presented significantly reduced body weights relative to respective controls (*Figure 2C*), and they manifested several unusual abnormalities, including blepharoptosis (ptosis, drooping eyelids) or microphthalmia (*Figure 2—figure supplement 1A and B*), abnormal posture

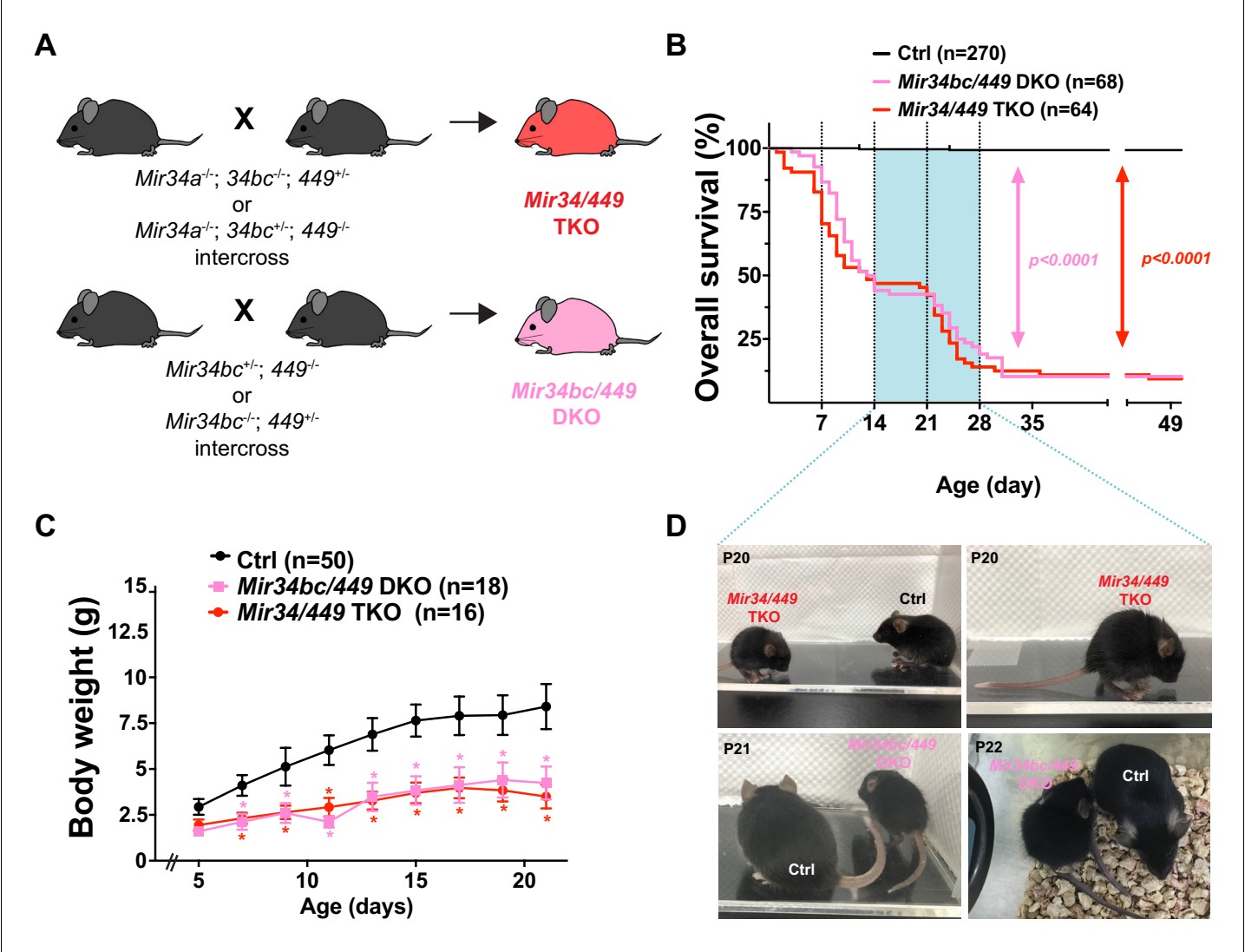

**Figure 2.** *Mir34bc/449* DKO and *Mir34/449* TKO mice exhibit marked postnatal lethality and survivors present profound neurological disorders. (**A**) Experimental strategy to generate the *Mir34/449* TKO or *Mir34bc/449* DKO mice. (**B**) Survival rates of *Mir34bc/449* DKO and *Mir34/449* TKO mice compared to littermates (Ctrl; all littermates of various genotypes have been pooled), and compared by log-rank test. Both *Mir34bc/449* DKO and *Mir34/449* TKO mice show similar mortality levels in the first 4 weeks. (**C**) *Mir34bc/449* DKO and *Mir34/449* TKO mice display reduced body weight compared to all Ctrl mice. Data are presented as mean ± SD. Two-way ANOVA with Tukey's multiple comparisons test (for each day); * denotes p<0.05. (**D**) Side and top views of *Mir34bc/449* DKO or *Mir34/449* TKO mice exhibiting abnormal body posture compared with the Ctrl, such as back-arching, curled body, hind tiptoe walking, and stimulus-induced hypersensitivity (see videos for details).

The online version of this article includes the following video and figure supplement(s) for figure 2:

**Figure supplement 1.** Blepharoptosis phenotype of MiR34/449-deficient mice.

**Figure 2—video 1.** Peculiar walking pattern in the MiR34/449 mutant mice.

https://elifesciences.org/articles/63768#fig2video1

**Figure 2—video 2.** Jumpy and rushed movement in the MiR34/449 mutant mice.

https://elifesciences.org/articles/63768#fig2video2

(back-arching; *Figure 2D*), and uncoordinated hind tiptoe walking and jumpy behavior, which resemble neurological deficits related to sensory-motor function defects (*Imai et al., 2016*; *Figure 2—videos 1* and *2*). Given that MiR34/449 miRNAs are highly expressed in the spinal cord (*Figure 1*) and our MiR34/449 mutant lines displayed apparent neurological disorders and peculiar moving patterns

(*Figure 2—videos 1* and *2*), we subsequently focused on identifying the functions of MiR34/449 miRNAs in mouse spinal cord.

## MiR34/449-deficient mice display normal spinal MN development with jumpy pattern

As both spinal INs and MNs exhibited strong MiR34/449 expression (*Figure 1N and O*), we first explored if MN differentiation is affected upon MiR34/449 depletion. Detailed examination of the ventral spinal progenitor domain by immunostaining with a battery of cross-repressive transcription factor hallmarks (*Chen and Chen, 2019*) did not uncover obvious progenitor domain changes in MiR34/449 mutant embryos at embryonic day E10.5 (*Figure 3—figure supplement 1A*).

Next, we investigated if MN differentiation was abrogated by *Mir34/449* locus deletion. We crossed our *Mir34/449* TKO mice with the MN reporter Hb9::GFP mouse line to facilitate MN identification. The number and distribution of post-mitotic MNs were comparable between MiR34/449-deficient and Ctrl embryos (*Figure 3—figure supplement 1B*). As our previous study showed that specific motor columns are preferentially affected and motor pools are eroded in conditional MN-Dicer KO mice (*Chen and Wichterle, 2012*), we assessed if MN subtype is affected in the *Mir34bc/449* DKO and *Mir34/449* TKO mice. To do that, we conducted immunostaining with the following known markers to identify each MN column type in specific spinal segments: MMC (medial motor column; $Lhx3^{on}$), LMC(m) (medial division of lateral motor column; $FoxP1^{on}$, $Isl1^{on}$), LMC(l) (lateral division of lateral motor column; $FoxP1^{on}$, $Isl1^{off}$), HMC (hypaxial motor column; $Lhx3^{off}$, $Isl1^{on}$), and PGC (preganglionic motor column; $FoxP1^{on}$, $pSmad^{on}$). However, MN subtype identity did not appear to be altered in MiR34/449-deficient embryos at E12.5 (*Figure 3—figure supplement 1C and D*). Thus, MiR34/449 miRNAs do not play an overt role in MN development.

Despite MN molecular identity not being affected by MiR34/449 deficiency, this does not necessarily mean that motor function at physiological or behavior levels is not disrupted. Therefore, we tested basal motor activity and locomotor coordination of mutant mouse lines by means of open field, rotarod, and treadmill tests (*Figure 3* and *Figure 3—figure supplement 2*). We also conducted these assays on age-matched WT mice to assist evaluations of the phenotypic variance caused by interbreeding complex genetic KO mice. Our results show that both *Mir34bc/449* DKO and *Mir34/449* TKO mice exhibited slightly yet significantly reduced locomotor activity relative to WT and Ctrl mice (*Figure 3B*), while their locomotor coordination appeared normal assayed by the rotarod (*Figure 3C and D*). Notably, we recorded obvious and spontaneous jumpy movements in the home cages of *Mir34bc/449* DKO and *Mir34/449* TKO mice, as well as in the open arena (occurrence ~35%; *Figure 2—video 2*). Ctrl mice with single or double combination KO with other MiR34/449 members also manifested this behavior with a milder extent, whereas the aged-matched WT hardly showed this phenomenon.

Subsequently, we performed treadmill to assay fine-scale kinematic study of limb coordination and gait patterns (*Herbin et al., 2007*; *Leblond et al., 2003*). Both *Mir34bc/449* DKO and *Mir34/449* TKO mice showed comparable locomotion patterns relative to WT and Ctrl mice in terms of running speed (limb moving speed) (*Figure 3E*) and temporal compositions during the step cycle (*Figure 3F and G*). Moreover, phase analyses of inter-limb coordination, including alternating homologous and homolateral limbs as well as synchronous diagonal limbs, were all normal (relative to WT and Ctrl) among MiR34/449 mutant mice during treadmill walking (*Figure 3H*), indicating that the ventral motor circuitry remained intact in the absence of MiR34/449. Finally, at the molecular level, immunostaining of $ChAT^{on}$ MNs in postnatal spinal cords further revealed that MiR34/449-deficient mice had similar numbers of MNs compared to Ctrl (*Figure 3—figure supplement 1E and F*). Based on these findings, we suggest that the overall motor function appeared hyper-jumpy while most motor outputs were largely preserved upon loss of MiR34/449 miRNAs.

## MiR34/449-deficient mice exhibit hypersensitivity in response to thermal stimulation

Since MiR34/449 also displayed strong expression in dorsal INs (*Figure 1O*), we examined if it is required to transduce pain modalities using several behavioral paradigms (*Figure 4A*; *Deuis et al., 2017*; *Gregory et al., 2013*). We first performed a von Frey test on MiR34/449 mutants, Ctrl and WT mice to measure their responses to mechanical stimulation, and observed that the response

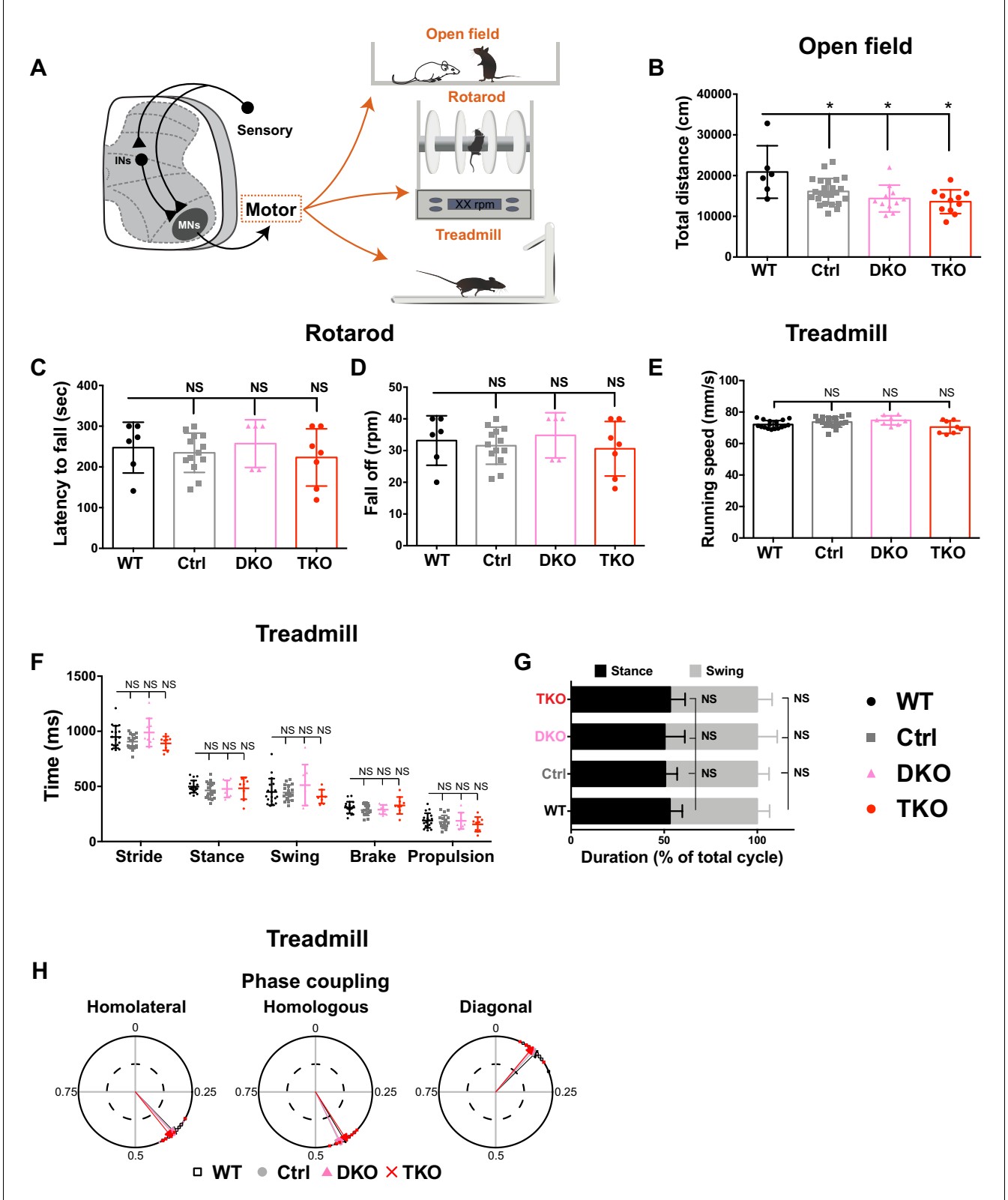

**Figure 3.** MiR34/449-deficient mice display overt jumpy pattern with no obvious deficit in basic motor function. (**A**) Schematic illustration of the behavioral tests conducted to assess motor function. (**B**) Quantification of locomotor activity in an open-field arena (assessed according to total distance traveled) for WT, Ctrl and indicated *Mir34/449* KO mice (WT: *n = 6*, Ctrl: *n = 24*, *Mir34bc/449* DKO: *n = 11*, *Mir34/449* TKO: *n = 11*). (**C and D**) Locomotor coordination on an accelerating rotarod, displayed as latency to fall (**C**) and the rotation speed at which mice fell off (**D**). There were no

*Figure 3 continued on next page*

*Figure 3 continued*

differences between MiR34/449 mutants and WT/Ctrl mice (WT: *n = 6*, Ctrl: *n = 13*, *Mir34bc/449* DKO: *n = 5*, *Mir34/449* TKO: *n = 7*). (E–H) Limb coordination and gait analyses were assayed by treadmill walking. The MiR34/449 mutants and WT/Ctrl mice have comparable running speeds (E), temporal parameters of the hindlimb (stride, stance, swing, brake, propulsion time; F), and ratios of swing and stance phases during the gait cycle (G). Limb coordination (homolateral, homologous, and diagonal phase coupling) is displayed as circular plots (H). Each symbol represents one trial from an individual mouse, and the arrow vectors represent the mean phases from all experiments for each group of genotyped mice. The length of the vector indicates the concentration of phase values around the mean (WT: *n = 18*, Ctrl: *n = 18*, *Mir34bc/449* DKO: *n = 8*, *Mir34/449* TKO: *n = 8*). No significant differences were observed among MiR34/449 mutant mice and Ctrl/WT mice. Data are presented as mean ± SD; one-way ANOVA with Tukey's multiple comparisons test; * denotes $p<0.05$; 'NS' denotes 'not significant'.

The online version of this article includes the following figure supplement(s) for figure 3:

**Figure supplement 1.** MN development may not be grossly affected in the spinal cord of *Mir34/449* KO mice.

**Figure supplement 2.** MiR34/449-deficient mice display subtle change of hindlimb moving pattern.

threshold in nociceptive withdrawal behavior was comparable among WT, Ctrl, and MiR34/449 mutant lines (*Figure 4B*).

Next, we measured responses to heat stimulation by means of a tail flick assay (representing a spinal reflex), whereby the tail withdrawal response is evaluated upon applying a heat stimulus to a mouse's tail (*Figure 4C*; *Chapman et al., 1985*; *Irwin et al., 1951*). We found that MiR34/449 mutant mice displayed a significantly reduced latency in the tail flick test relative to both Ctrl and WT mice (*Figure 4C*), indicative of increased sensitivity to heat stimulation. We then performed a hot-plate experiment whereby a heat stimulus is applied to the hind paws of mice, as this assay is considered to involve more complex supraspinal pathways (*Giglio et al., 2006*). We observed a minor (but not statistically significant) decrease in hindlimb response latency among MiR34/449 mutant mice when compared to WT and Ctrl mice (*Figure 4D*). Thus, we suggest that the major function of MiR34/449 in the spinal cord relates to spinal reflex circuits.

## Satb1 and Satb2 are direct targets of MiR34/449 miRNAs in the spinal cord

Thereafter, we sought to identify the primary targets of MiR34/449 that might be involved in the prominent defects of spinal reflex circuits upon MiR34/449 deletions, first we catalogued *in silico*-predicted targets of MiR34/449 and performed gene ontology analysis using the DAVID online database (*Huang et al., 2009*; *Tung et al., 2015*). Interestingly, 'synapse organization', 'neurogenesis', 'transcription factor regulation', and 'regulation of ion transport' were the most prominent MiR34/449-mediated pathways arising from that analysis (*Figure 5A*). To establish which genes could explain the spinal reflex phenotype, we further filtered our predicted targets according to (1) whether they manifested both miRNA(MiR34/449-5p)/miRNA*(MiR34a-3p and MiR34b/c-3p) targeting sites and (2) exhibited strong IN expression in postnatal spinal cords (*Figure 5B*; *Lai et al., 2016*; *Osseward and Pfaff, 2019*). Based on those criteria, we identified potential candidate targets of MiR34/449 that were predicted to participate in IN specification or maturation, including multiple genes previously characterized as being important in INs, that is, *Satb1, Satb2* (*Figure 5B* and *Supplementary file 1g*). To determine if *Satb1/2* are direct targets of MiR34/449, we constructed luciferase reporters containing the full-length 3' UTR of those genes that harbor predicted MiR34/449 target sites (*Figure 5—figure supplement 1*). The miRNA target-predicting algorithms identified potential binding sites for MiR34/449-5p and MiR34/449-3p in the 3' UTR of these genes (*Figure 5—figure supplement 1*). Co-transfection of the luciferase construct with three MiR34/449 expression vectors resulted in a reduction in luciferase activity for the WT 3' UTRs of *Satb1/2* constructs (*Figure 5C and D*), whereas a construct harboring the mutated targeting sequences was completely insensitive to miRNA-mediated silencing (*Figure 5C and D*). These findings indicate that MiR34/449 targets the 3' UTRs of *Satb1* and *Satb2*.

Notably, a previous study characterized the cellular identity of MSE neurons and found that *Satb1/2* are expressed in medial lamina V/VI and lamina III (*Levine et al., 2014*). Unlike for WT and control littermates at P14, we observed a significantly increased number of Satb1[on] and Satb2[on] INs in the MiR34/449 mutant spinal cords (*Figure 5E~5G*). Moreover, we also observed a drastic increase in the number of hybrid cells expressing both Satb1 and Satb2 in the MiR34/449 mutant

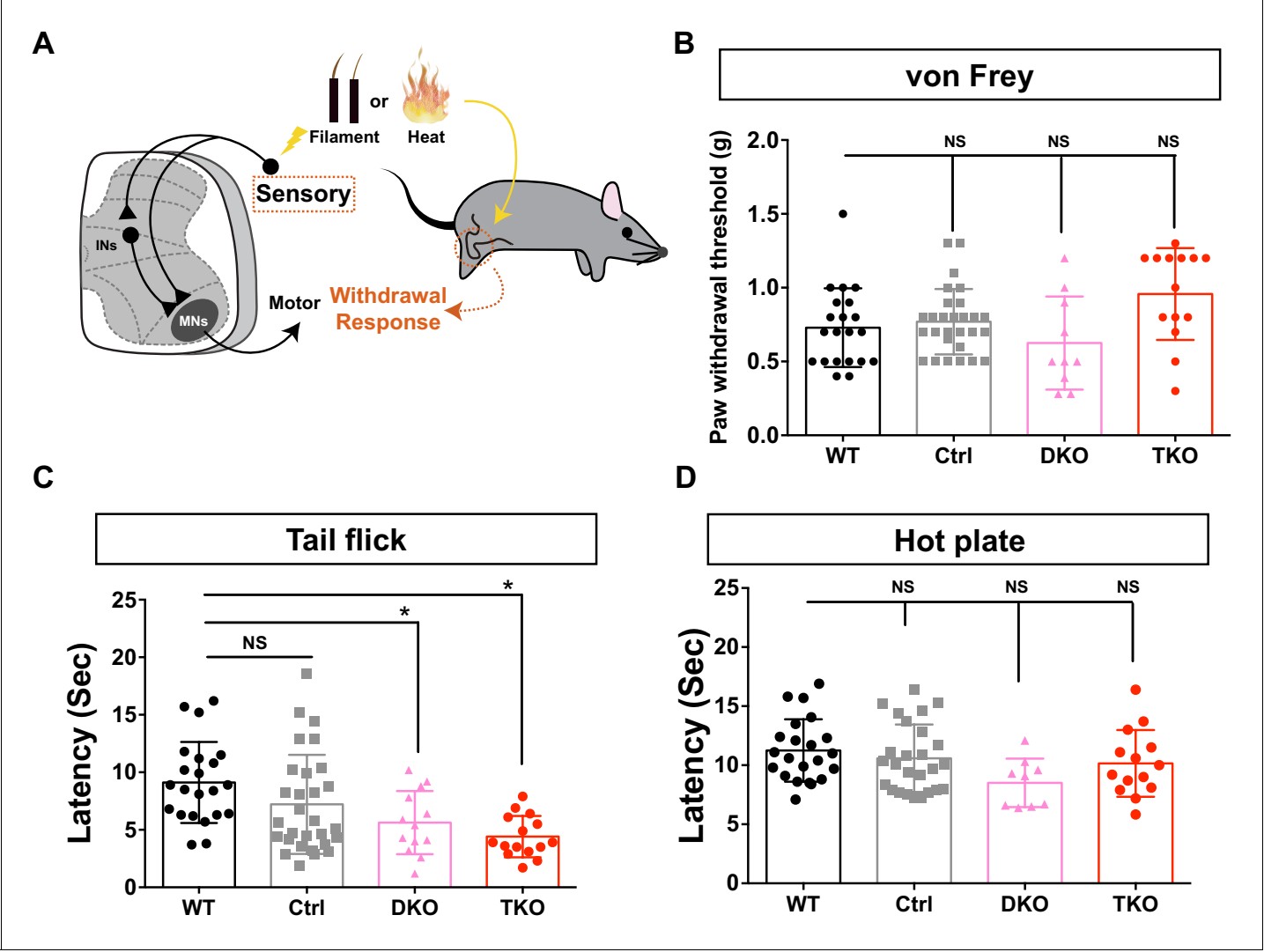

**Figure 4.** *Mir34/449* KO mice exhibit hypersensitivity in response to thermal stimulation. (**A**) Schematic illustration of the sensory assays. (**B**) A von Frey test revealed comparable thresholds for inducing a mechanical pain response among 4-month-old WT, Ctrl, and MiR34/449 mutant mice (WT: *n = 21*, Ctrl: *n = 28*, *Mir34bc/449* DKO: *n = 10*, *Mir34/449* TKO: *n = 14*). Data are represented as mean threshold (grams of force) ± SD. (**C**) MiR34/449 mutant mice exhibit greater heat pain sensitivity compared to WT in a tail-flick test. Ctrl mice also showed a reduced latency in response to the heat stimulus relative to WT, but it is not statistically significant (WT: *n = 22*, Ctrl: *n = 29*, *Mir34bc/449* DKO: *n = 13*, *Mir34/449* TKO: *n = 15*). (**D**) Heat thresholds to induce a thermal pain response (hot-plate set at 55°C) are presented as the latency to licking/flicking the hind paw or jumping behavior. The MiR34/449 mutant mice display a subtle but not statistically significant difference in heat sensitivity compared to WT/Ctrl (WT: *n = 22*, Ctrl: *n = 26*, *Mir34bc/449* DKO: *n = 9*, *Mir34/449* TKO: *n = 14*). Data are presented as mean ± SD; one-way ANOVA with Tukey's multiple comparisons test; * denotes $p<0.05$; 'NS' denotes 'not significant'.

spinal cords at P14 compared to WT and Ctrl groups (*Figure 5E and H*). Thus, we have shown that Satb1 and Satb2 are directly regulated by MiR34/449 miRNAs in the spinal cord.

To further explore MiR34/449 and Satb1/2 expression *in vivo*, we conducted *in situ* hybridization of MiR34a-5p/MiR34c-5p and compared their expression patterns to that of Satb1/2. Notably, we identified many Satb1/2[on] INs that co-expressed MiR34a-5p or MiR34c-5p in the dorsal/intermediate regions of the spinal cord at P14, suggesting potentially direct regulation of Satb1/2 by MiR34/449 in spinal INs (*Figure 5—figure supplement 2*). Together, these findings indicate that MiR34/449 may control optimal development of Satb1/2[on] INs in the spinal cord to fine-tune sensory-motor circuit outputs.

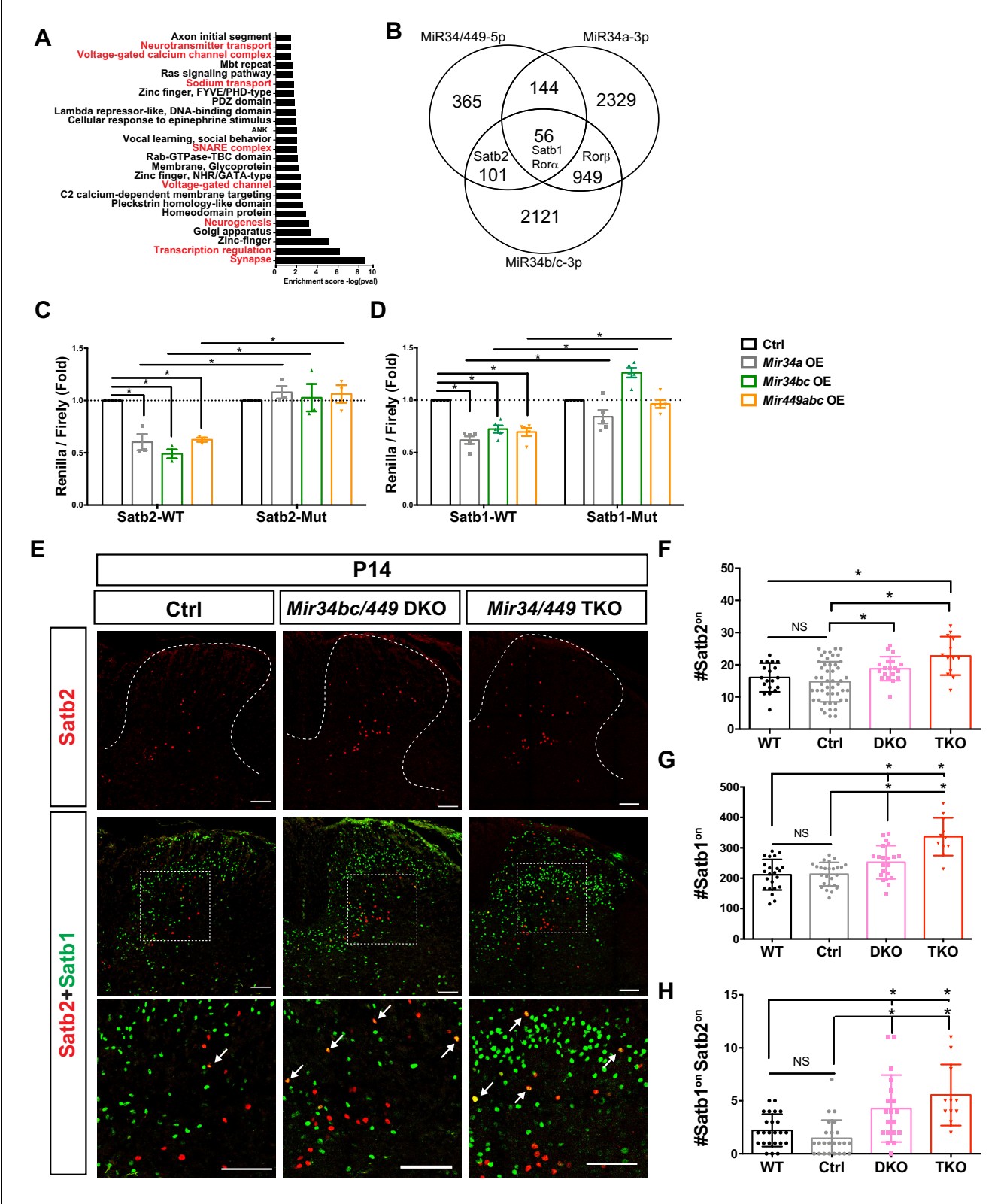

**Figure 5.** Satb1 and Satb2 are direct targets of MiR34/449 miRNAs. (**A**) Strategy to identify potential MiR34/449 targets in the spinal cord. Predicted targets of MiR34/449 miRNAs by TargetScan were subjected to gene ontology analysis. Potential pathways related to the neural behavior phenotype are highlight with red color (**B**) Predicted targets of MiR34/449 miRNAs were filtered based on having both miRNA (MiR34/449-5p) and miRNA* (MiR34b/c-3p and MiR34a-3p) targeting sites, and spinal INs were selected for further study. (**C–D**) Co-expression of the luciferase construct with

*Figure 5 continued on next page*

*Figure 5 continued*

MiR34/449 plasmids in HEK293T cells silences the reporter carrying the intact MiR34/449 target sites, whereas *Mir34/449* plasmids fail to silence Satb2-Mut (C) and Satb1-Mut (D) luciferase constructs. Transfection of the backbone plasmid served as the control (Ctrl). OE, overexpression. *N = 3* in (C) and *five* in (D). (E) Immunostainings for Satb2 (*red*) and Satb1 (*green*) in the dorsal and intermediate spinal cord of cervical/brachial sections at P14 revealed significantly increased numbers of Satb2$^{on}$ and Satb1$^{on}$ cells. Lower panels are higher magnifications of the boxed areas in the respective middle panels. Double-positive cells are indicated with arrows. (F–H) Quantification of Satb2$^{on}$ (F), Satb1$^{on}$ (G), and Satb2$^{on}$;Satb1$^{on}$ double-positive INs (H) from (E). Each dot represents the quantification from one hemi-section. Data are presented as mean ± SD. One-way ANOVA with Tukey's multiple comparisons test. * denotes $p<0.05$; 'NS' denotes 'not significant'. Scale bars represent 100 µm.

The online version of this article includes the following figure supplement(s) for figure 5:

**Figure supplement 1.** Luciferase reporter and mutant construction.

**Figure supplement 2.** Expression of MiR34a-5p and MiR34c-5p in the postnatal spinal cord.

**Figure supplement 3.** Spinal IN development may not be grossly affected in the MiR34/449 mutant mice.

**Figure supplement 4.** Numbers of Satb2$^{on}$ INs are increased in both the thoracic and lumbar regions of the spinal cord.

**Figure supplement 5.** Spatial distribution of Satb2$^{on}$ INs.

## MiR34/449 tunes optimal postnatal Satb2 expression in Ctip2 subpopulation

While the function of Satb1 in the spinal interneurons is not elucidated yet, a previous study revealed that Satb2 expression is important for establishing appropriate local connectivity in spinal circuits (*Hilde et al., 2016*), highlighting the possibility that MiR34/449-regulated Satb2 expression might be involved in IN development and consequently contribute to sensory-motor circuit outputs. To test this possibility, we first examined if IN development is affected upon MiR34/449 abrogation. At E11.5, there were no apparent differences in IN progenitors in the dorsal or ventral horns—as revealed by molecular markers Lbx1 (dI4 ~6), Lhx1/5 (dI2, 4, 6~V1), Isl1 (dI3), and Tlx3 (dI3, 5) for the dorsal horn, or Foxd3 (dI2, V1), Chx10 (V2a), and Lhx3 (V2) for the ventral horn, respectively—indicating that MiR34/449 is not required for most IN development (*Figure 5—figure supplement 3A*~K). At E13.5 (the earliest time-point at which we could detect dorsal Satb2 INs), there was no significant difference in numbers of Satb2$^{on}$ INs in MiR34/449 mutant spinal cords relative to WT and Ctrl (*Figure 5—figure supplement 3L and M*). Although comparable numbers of Satb2$^{on}$ INs persisted until E16.5 (data not shown), there was a dramatic increase of Satb2$^{on}$ INs relative to WT and Ctrl at P14 in the MiR34/449 mutant spinal cords throughout the spinal cord segments (*Figure 5E and F* and *Figure 5—figure supplement 4*). This outcome indicates that optimal Satb2$^{on}$ IN number and maintenance is particularly sensitive to MiR34/449 expression in the postnatal spinal cord. Collectively, we suggest that MiR34/449 miRNAs control optimal development of Satb2-expressing INs in the spinal cord, particularly at the postnatal stage, enabling this miRNA family to fine-tune and maintain sensory-motor circuit outputs.

Given that the position of the Satb2$^{on}$ INs is highly associated with the connectivity of the sensory afferents (*Hilde et al., 2016*; *Levine et al., 2014*), we next tested if the position of the Satb1$^{on}$ or Satb2$^{on}$ INs is affected in the absence of the MiR34/449 at P14. However, neither Satb1 or Satb2 expressing IN positions were obviously affected (*Figure 5—figure supplement 5*), reflecting the different phenotype manifestation between MiR34/449 mutants and Satb2 KO mice (*Hilde et al., 2016*; *Levine et al., 2014*).

## Altered Satb2$^{on}$ IN subpopulations in the MiR34/449 mutant spinal cord

Based on previous study (*Hilde et al., 2016*), the molecular identity of the spinal cord is altered upon loss of Satb2, particularly with regard to Ctip2- and Pax2-expressing IN subpopulations at the embryonic stage. Thus, we further examined if the increased Satb2$^{on}$ INs in the MiR34/449 mutant spinal cords display an altered complement of subpopulations. Notably, we observed an increase in the number of Satb2$^{on}$;Ctip2$^{on}$ INs in the MiR34/449 mutant spinal cords (*Figure 6A and B*), yet only a subtle (but not statistically significant) change in the Satb2$^{on}$;Pax2$^{on}$ IN subpopulation was found (*Figure 6A and C*). As the individual Ctip2$^{on}$ and Pax2$^{on}$ cell populations were of comparable size, this outcome further indicates that the Satb2$^{on}$;Ctip2$^{on}$ IN subpopulation had preferentially increased, with a concomitant decrease in the Satb2$^{on}$;Pax2$^{on}$ subpopulation, in the *Mir34/449* TKO spinal cords relative to those of WT and control littermates (*Figure 6D and G*). Further spatial distribution analyses revealed no significant change in the distribution of the increased number of

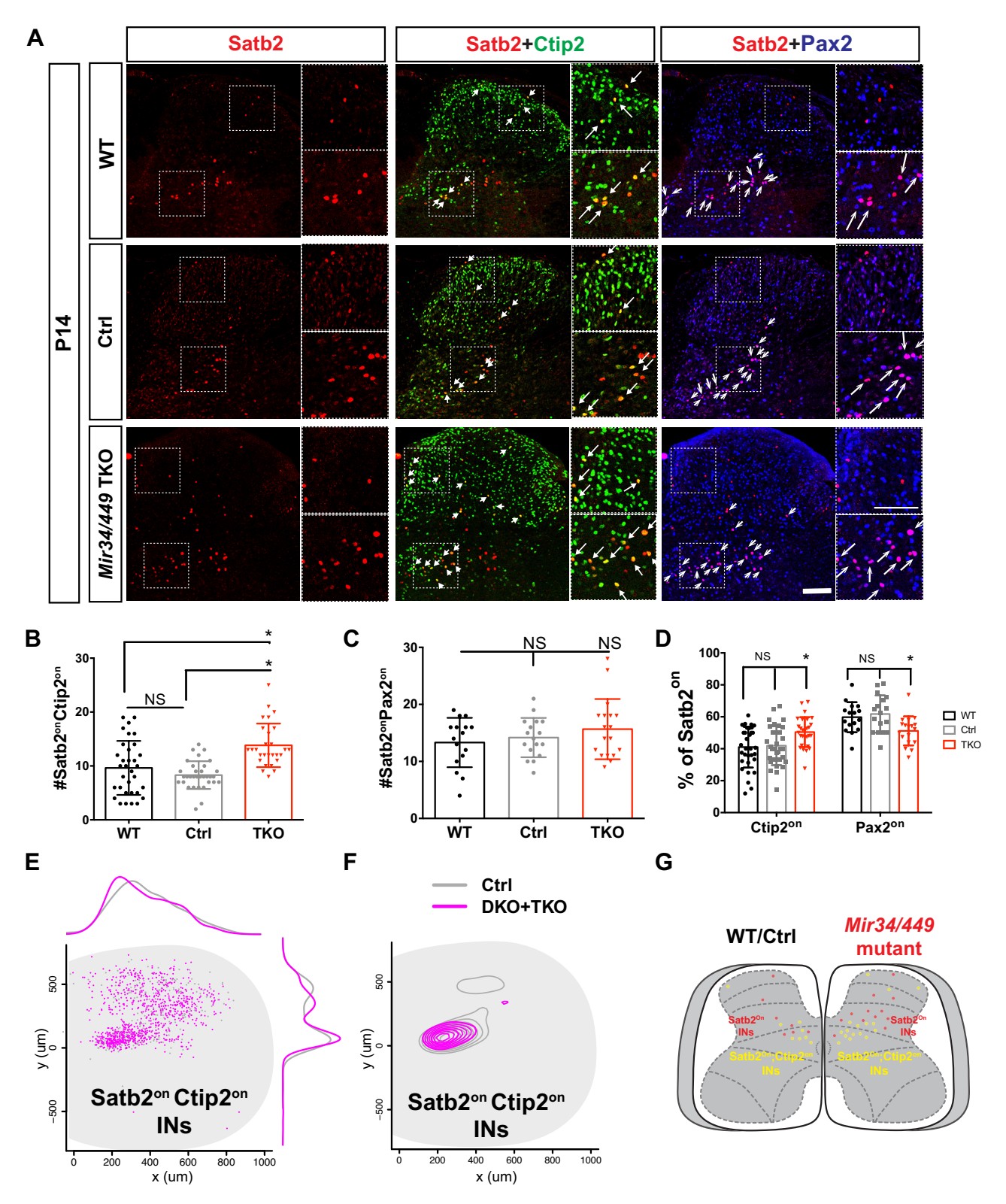

**Figure 6.** MiR34/449 tunes optimal Satb2 expression in the Ctip2 IN subpopulation. (**A**) Immunostaining for Satb2, Ctip2, and Pax2 in dorsal and intermediate spinal cord at P14. Note that Satb2$^{on}$ INs display two molecular identities, i.e., the Ctip2$^{on}$ or Pax2$^{on}$ subpopulations. Comparison with WT/Ctrl reveals an increase in the cell number of Satb2$^{on}$;Ctip2$^{on}$ INs in the intermediate spinal cord of cervical/brachial sections upon MiR34/449 deletion. Boxed areas indicate higher magnification regions from the intermediate and dorsal regions of each image. Arrows highlight the presence of

*Figure 6 continued on next page*

*Figure 6 continued*

dual-labeled Satb2on;Ctip2on cells (*yellow*) or dual-labeled Satb2on;Pax2on cells (*magenta*) among the Satb2on population. (B–C) Quantifications of cell number from (A). (D) Quantifications of the Satb2on;Ctip2on and Satb2on;Pax2on subpopulations were normalized against total Satb2on cell numbers and are shown as a percentage. (E) Comparison of the spatial distribution of individual Satb2on;Ctip2on INs in the cervical/brachial spinal cord of indicated mice (Ctrl, *grey*; *Mir34bc/449* DKO and *Mir34/449* TKO, *magenta*). Frequency distributions along the medio-lateral (*top*) and dorso-ventral (*right*) axes represent non-linear regressions. (F) A contour density plot of Satb2on;Ctip2on INs indicates that Satb2on;Ctip2on INs are mainly located in the medial spinal cord of both MiR34/449 mutant and Ctrl mice. Contour lines represent density at the 30th–90th percentiles. Data were analyzed from more than 25 sections of at least five independent spinal cords in (E) and (F). (G) Schematic illustrating the increased number of Satb2on;Ctip2on INs in the MiR34/449 mutant spinal cord. Data are presented as mean ± SD. One-way ANOVA with Tukey's multiple comparisons test. * denotes p<0.05; 'NS' denotes 'not significant'. Scale bars represent 100 µm.

Satb2on;Ctip2on, yet these cells tended to exhibit a more condensed pattern in the intermediate region of the MiR34/449 mutant spinal cord (*Figure 6E and F*). Given all of these findings, we suggest that MiR34/449 controls optimal development of Satb2-expressing INs in the spinal cord, and that loss of MiR34/449 leads to a change in the molecular identity of Satb2on IN subtypes. Our results reflect a potential role for these miRNAs in maintaining precise cell identity in the postnatal spinal cord, which is critical for refining sensory-motor circuit outputs (*Figure 7*).

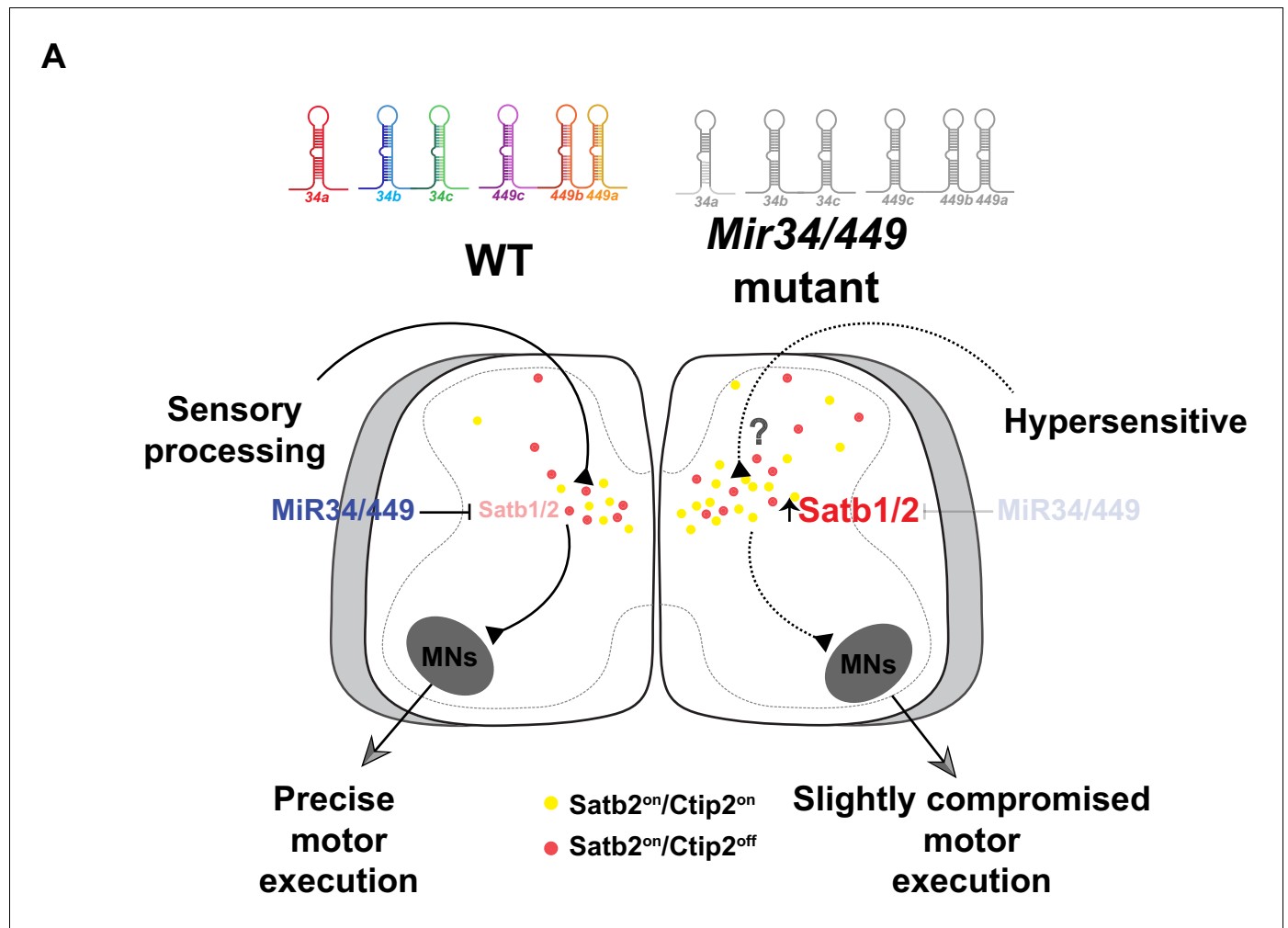

**Figure 7.** MiR34/449 miRNAs mediate control of proper interneuron populations to exert precise core sensory-to-motor spinal outputs. Summary diagram of the increased numbers of Satb1/2on INs, and specifically the Satb2 on;Ctip2on subpopulation, in MiR34/449 mutant mice and the possible changes in circuitry that result from MiR34/449 depletion in the spinal cord.

## Discussion

A hallmark of miRNAs is that they are often expressed in a cell type-specific manner at various developmental stages, allowing them to modulate a variety of biological processes such as cell survival, differentiation, and cell fate transitions (*Conaco et al., 2006*; *Shenoy and Blelloch, 2014*; *Tuncdemir et al., 2015*). The MiR34 family is believed to be regulated by p53 in tumor cell contexts, endowing them with roles in cell proliferation and p53-mediated apoptosis (*Bommer et al., 2007*; *Choi et al., 2011*; *Corney et al., 2007*; *He et al., 2007*; *Okada et al., 2014*; *Raver-Shapira et al., 2007*). Recent studies in which all three mouse *Mir34/449* alleles were completely deleted pointed to unexpected functions of miRNAs of MiR34/449 family in motile ciliogenesis of multiciliated epithelial cells of the respiratory and reproductive systems, perhaps reflecting a p53-independent mechanism (*Otto et al., 2017*; *Song et al., 2014*; *Wu et al., 2014*). Although the MiR34/449 family is strongly expressed in the CNS, their functions there had remained surprisingly unexplored. Here, we reveal several unusual phenotypes of MiR34/449 mutant mice at postnatal stages (P14~P28), including blepharoptosis (drooping eyelids) or microphthalmia, abnormal posture with back-arching, hind tiptoe walking, jumpy behavior, and stimulant-induced hypersensitivity. We have demonstrated that MiR34/449 miRNAs target Satb1/2 to fine-tune Satb1/2$^{on}$ IN numbers in the spinal cord. Depletion of MiR34/449 resulted in an increased subpopulation of Satb2$^{on}$;Ctip2$^{on}$ INs, which might contribute to some of the above-mentioned phenotypes (*Figure 7*). Our study highlights an unappreciated role of MiR34/449 in the postnatal CNS, and we discuss the implications of these results further below.

### Role of miRNAs in spinal interneurons

The roles of miRNAs during spinal MN development have been relatively well illustrated. Specifically, we have previously shown that MiR17-3p participates in the Olig2/Irx3 bistable loop to carve out the pMN/p2 boundary, and MiR27 operates in the Hoxa5/Hoxc8 regulatory circuit to control the timing of Hoxa5 protein expression to exert brachial MN pool identity (*Chen et al., 2011*; *Li et al., 2017*). However, despite a series of elegant experiments demonstrating spinal IN identity at molecular and physiological levels (*Levine et al., 2014*), the functions of miRNAs in spinal INs and their relationships to the transcription factor-mediated loop have remained puzzling. Here, we have demonstrated that the MiR34/449 family directly regulates Satb1/2 expression at the late differentiation stage, thereby potentially altering sensory-motor circuits given that Satb1/2$^{on}$ INs act as a hub located in the intermediate part of the spinal cord. As Satb1/2$^{on}$ INs receive inputs from multiple streams of sensory information and relay their outputs to motor command layers of the spinal cord (*Hilde et al., 2016*), it is conceivable that our *Mir34/449* KO mice exhibited hypersensitivity to these sensory inputs, accounting for the peculiar walking patterns we observed. Of the IN types in the dorsal spinal cord, why does Satb1/2 expression and the number of Satb1/2$^{on}$ INs need to be precisely controlled by MiR34/449? Perhaps motor behaviors such as walking or pain-induced limb withdrawal depend on integration of multimodal sensory circuits to elicit appropriate responsive muscle contractions. Moreover, Satb1/2$^{on}$ INs are particularly critical in interpreting multiple sensory streams and coordinating them into singular behavioral outputs, with loss of Satb2 leading to impairments of IN cellular position, molecular profile, and pre- and post-synaptic connectivity (*Hilde et al., 2016*). The complex behavioral repertoire of animals is regulated by a set of motor synergy encoder (MSE) neurons, in which both Satb1 and Satb2 are prominent molecular regulators (*Levine et al., 2014*). In contrast to Satb2, the function of Satb1$^{on}$ INs is not yet completely understood (*Hilde et al., 2016*). Although Satb1 and Satb2 are close homologs, they may regulate genes via convergent and divergent pathways. In this study, we have also uncovered that Satb1 and Satb2 are expressed largely exclusively in spinal INs. Upon MiR34/449 loss-of-function, both Satb1 and Satb2 are aberrantly upregulated, with a consequent increase in Satb1/2 co-expressing cells. This scenario highlights the potential involvement of miRNAs in the control of mediating intricate balance of the MSE neurons formation, as well as the critical involvement of miRNAs in maintaining spinal neural circuits at the postnatal stage (*Imai et al., 2016*).

We found that expression of *Satb2* is aberrantly upregulated in our MiR34/449 mutant mice. Moreover, we revealed the Satb2$^{on}$;Ctip2$^{on}$ subpopulation change upon MiR34/449 deletion which enhancement may result in the aberrant connection in the spinal circuitry. Therefore, similar to miRNA-mediated establishment of MN subtype identity (*Tung et al., 2015*), we suggest that a Satb1/2-MiR34/449 regulatory axis might be critical to exert and fine-tune the precise level of

*Satb1/2* expression and to refine Satb1/2$^{on}$ IN number for receipt of multimodal sensory inputs. This latter is pivotal for animals to react appropriately to diverse sensory stimuli (*Figure 7*). In cortical neurons, Satb2 inhibits Ctip2 activity to define two major classes of projection neurons (*Alcamo et al., 2008*; *Britanova et al., 2008*). However, some neurons in perinatal mice manifest a subpopulation of Satb2/Ctip2 co-expressing cortical neurons (*Harb et al., 2016*). Interestingly, post-natal Satb2$^{on}$ spinal INs also co-express Ctip2 at the medial region. While the genetic network regulating this Satb2/Ctip2 subpopulation in the cortex and spinal cord remains enigmatic, it is clear that the considerable diversity of projection neurons and spinal INs in mammals cannot simply be attributable to Satb2 inhibitory activity. Consequently, the regulatory mechanism underlying Satb2$^{on}$;Ctip2$^{on}$ neuron subtypes has yet to be identified. In this study, we have unveiled that numbers of Satb2$^{on}$;Ctip2$^{on}$ spinal INs increase upon *Mir34/449* KO in mice, raising the possibility that one major function of MiR34/449 is to refine the optimal MSE neurons for precise sensory-motor outputs. Given the prominent activity-dependent and tuning role of miRNAs (*Chen and Chen, 2019*; *Sim et al., 2014*), it is tantalizing to test in the future if MiR34/449 also establishes the balance of Satb2$^{on}$;Ctip2$^{on}$ INs among cortical projection neurons.

Given that MiR34/449 also targets a series of synaptic genes (*Figure 5A*), it warrants further investigation if MiR34/449 also engages in the pre- and post-synaptic connectivity by Satb1/2$^{on}$ INs to support input convergence. Importantly, other IN genes such as *Rorα* and *Rorβ* are potential targets of MiR34/449 (*Figure 5B*), so whether MiR34/449 miRNAs regulate a battery of spinal INs and if disruption of those pathways account for diverse neurological disorders are interesting topics for further study. Given that Dicer is required to maintain monosynaptic sensory-motor circuits in the spinal cord (*Imai et al., 2016*), it is conceivable to hypothesize that other miRNA candidates might be involved in establishing spinal IN subtype identity and regulating sensory-motor connectivity in the spinal cord. Experiments to elucidate the full spectrum of miRNAs involved in these mechanisms will provide deep insights into spinal circuit formation and control.

## Redundant and non-redundant functions of MiR34/449 in the spinal cord

A series of studies have revealed that removal of individual miRNA paralogs alone results in partial penetrance of phenotypic deficits. For instance, *Mir196* TKO gives rise to homeotic patterning defects, whereas single/compound *Mir196* KO mice exhibited similar defects to varying degrees (*Wong et al., 2015*). In agreement with previous studies (*Bao et al., 2012*; *Concepcion et al., 2012*; *Song et al., 2014*), we did not observe an overt phenotype in our *Mir34* or *Mir449* single KO lines relative to Ctrl mice. However, a hypersensitivity stress test revealed that our Ctrl mice harboring at least one *Mir34/449* allele exhibited trivial but statistically non-significant hypersensitivity in response to a heat stimulus (*Figure 4C*). Hence, it was important that we also subjected age-matched WT mice from an identical genetic background to our assays for comparative analyses (*Andolina et al., 2018*; *Andolina et al., 2016*; *Otto et al., 2017*; *Wu et al., 2014*). To obtain reasonable numbers of *Mir34bc/449* DKO and *Mir34/449* TKO mice for our experiments, we intercrossed the mice possessing one WT *Mir34/449* allele (*Figure 2A*; see Materials and methods). Therefore, our molecular and behavioral characterizations reflect comparisons between *Mir34bc/449* DKO or *Mir34/449* TKO mice and both WT and Ctrl lines, with the genetically-matched Ctrl mice displaying complex heterogeneity and possessing at least one and up to at most five alleles targeted for deletion in KO lines (*Simpson et al., 1997*). Our observations provide evidence for two roles of miRNAs during development: (1) miRNAs can function redundantly to control or fine-tune critical developmental steps, and (2) overt phenotypes upon disrupting only some miRNA family members might only be evident at physiological levels when stress conditions are encountered (*Bartel, 2018*; *Chen and Chen, 2019*).

Why is it that *Mir34bc/449* DKO mice displayed a seemingly similar defective phenotype to *Mir34/449* TKO mice, whereas the *Mir34a/449* DKO mice are largely normal? Given that MiR34a seems to be expressed at high levels ubiquitously in almost all cell types, including spinal neurons (*Concepcion et al., 2012*; *Otto et al., 2017*; *Song et al., 2014*; *Wu et al., 2014*), this is a puzzling scenario. It is generally accepted that pre-microRNA is processed through Dicer to generate a miRNA duplex, consisting of miRNA and miRNA* strands (also termed guide and passenger strands, respectively). Although most studies assume that miRNA* has no regulatory activity, some evidence exists to suggest that the abundance, functionality, and physiological relevance of miRNA* are underestimated (*Okamura et al., 2008*). Our previous study showed that both MiR17-5p (guide

strand) and MiR17-3p (passenger strand, i.e. MiR17*) are equally functionally important in precursor MN patterning and MN differentiation, having the same or different targets (*Chen et al., 2011*). Interestingly, in the current study, we found that MiR34b-3p (MiR34b*) is significantly expressed in the developing spinal cord (data not shown), raising the possibility that MiR34b-3p might also be functionally relevant, even though its seed sequence is not conserved when compared to that of MiR34a-3p (MiR34a*).

We focused on Satb1 and Satb2 in this study as it is a shared target of both MiR34a-5p (seed, GGCAGUG) and MiR34b/c-3p (seed, AUCACUA) (*Figure 5—figure supplement 1A*). Accordingly, we argue that one plausible reason why *Mir34bc/449* DKO mice exhibit a similar lethality phenotype to *Mir34/449* TKO mice, whereas *Mir34a/449* DKO mice are relatively normal, might reflect an unappreciated function of MiR34b/c-3p. We are now in the process of dissecting the possible convergent and divergent targets of the MiR34/449 miRNA family and their implications for embryonic development by examining individual *Mir34/449* KO embryonic stem cell lines in our possession.

## MiR34/449 miRNAs as therapeutic targets for tumors and CNS disease

The two most prominent and documented roles of MiR34/449 miRNAs are their tumor suppressor function in cancer cells (*He et al., 2007*) and their regulation of motile cilia in multiciliated cells (MCCs) during development (*Song et al., 2014*). From a tumor cell perspective, numerous cell- and animal-based studies have explored the tumor-suppressive function of MiR34a and proposed that restoring MiR34a expression could serve as a potential therapeutic anti-cancer strategy. An advantage of such MiR34a-based therapy is the possibility of simultaneously repressing multiple oncogenic and immune evasion pathways (*Di Martino et al., 2012*). Moreover, MiR34a overexpression does not seem to be excessively toxic to normal cells *in vitro* or *in vivo* (*Raver-Shapira et al., 2007*; *Tazawa et al., 2007*).

In terms of development, MiR34/449 is known to regulate ciliogenesis in MCCs that have hundreds of motile cilia projecting from their apical surfaces. The critical role of MiR34/449 in regulating MCCs is evolutionary conserved. Ciliation defects caused by MiR34/449 deficiency have been reported for MCCs of the murine airway and fallopian tube, the embryonic epidermis of *Xenopus*, and the multiciliated pronephros and nasal pits of zebrafish embryos (*Chevalier et al., 2015*). In addition to Satb2-mediated hypersensitivity to sensory stimuli in our MiR34/449 mutant mice, we uncovered a series of neurological disorders—particularly staggering, ataxia, and seizure-like behaviors—that could not be fully accounted for by aberrant upregulation of *Satb2* in the spinal cord. Ependymal cells in the nervous system extend their motile cilia into the brain ventricles and contribute to cerebrospinal fluid (CSF) flow, a process important for normal brain function and adult neurogenesis (*Silva-Vargas et al., 2016*). A previous study has shown that MiR449 miRNAs express in the early development of choroid plexus, supporting the putative roles of MiR34/449 in the CSF production (*Redshaw et al., 2009*). It is not yet clear if some of the neurological disorders we uncovered in this study reflect a defect in cilial motility or CSF secretion in the spinal cord. Given the prominent and conserved roles of MiR34/449 miRNAs in motile cilia, it will be tempting to analyze the relevance of ependymal cilia motility for CSF circulation and brain ventricle morphogenesis in conditional neural *Mir34/449* KO mice in the future.

In addition, the observed neurological defects in *Mir34/449* mutant mice may also have arisen from proprioceptive impairment since previous study has revealed that Satb2[on] INs reside in the terminal region of proprioceptive afferents (*Hilde et al., 2016*). As one set of measures of proprioceptive function, the spatial parameters of stride length we measured for the treadmill assay presented a trivial but statistically significant difference in MiR34/449-deficient mice relative to WT (*Figure 3—figure supplement 2E*), implying a potential involvement of the sensory connectivity function of MiR34/449 in the spinal cord. Although we have reported an increase in the number and dense positioning of the Satb2[on];Ctip2[on] IN subpopulation upon MiR34/449 abrogation, further assessments are required to validate a link between the proprioceptive afferents displaying this molecular change in IN identity and the peculiar mouse behaviors we observed.

Taken together, our findings reveal that MiR34/449 miRNAs have important yet underappreciated roles in regulating sensory-motor circuit outputs by refining IN numbers. As MiR34 has already undergone clinical trials as a potential gene therapy for cancer and we have uncovered several neurological functions for the MiR34/449 family, we envision that MiR34/449 miRNAs might serve as targets for treatments of neurodegenerative disease in the future.

# Materials and methods

## Key resources table

| Reagent type (species) or resource | Designation | Source or reference | Identifiers | Additional information |
|---|---|---|---|---|
| Antibody | Rabbit monoclonal anti-Satb2 | Abcam | Cat# ab92446, RRID:AB_10563678 | IHC (1:500) |
| Antibody | Guinea pig polyclonal anti-Satb2 | Synaptic Systems | Cat# 327004, RRID:AB_2620070 | IHC (1:500) |
| Antibody | Goat polyclonal anti-ChAT | Millipore/Sigma | Cat# AB144P, RRID:AB_2079751 | IHC (1:100) |
| Antibody | Sheep anti-GFP | Bio-Rad | Cat# 4745–1051, RRID:AB_619712 | IHC (1:2000) |
| Antibody | Mouse monoclonal anti-NeuN | Millipore | Cat# MAB377, RRID:AB_2298772 | IHC (1:500) |
| Antibody | Rabbit polyclonal anti-Irx3 | T. Jessell (Columbia, New York) | N/A | IHC (1:5000) |
| Antibody | Rabbit polyclonal anti-pSmad1/5/8 | Merck | Cat# AB3848, RRID:AB_177439 | IHC (1:500) |
| Antibody | Guinea pig anti-Olig2 | T. Jessell (Columbia, New York) | N/A | IHC (1:10000) |
| Antibody | Rabbit polyclonal anti-Olig2 | Millipore | Cat# AB9610; RRID:AB_570666 | IHC (1:5000) |
| Antibody | Mouse monoclonal anti-Nkx2.2 | T. Jessell (Columbia, New York) | N/A | IHC (1:10000) |
| Antibody | Guinea pig anti-Hb9 | H. Wichterle (Columbia, New York) | N/A | IHC (1:1000) |
| Antibody | Rabbit polyclonal anti-Lhx3 | Abcam | Cat# ab14555, RRID:AB_301332 | IHC (1:2000) |
| Antibody | Rabbit polyclonal anti-Foxp1 | Abcam | Cat# ab16645, RRID:AB_732428 | IHC (1:10000) |
| Antibody | Guinea pig anti-Foxp1 | T. Jessell (Columbia, New York) | N/A | IHC (1:20000) |
| Antibody | Mouse monoclonal anti-Isl1/2 | DSHB | Cat# 39.4D5, RRID:AB_2314683 | IHC (1:1000) |
| Antibody | Goat polyclonal anti-Isl1 | Neuromics | Cat# GT15051, RRID:AB_2126323 | IHC (1:100) |
| Antibody | Mouse monoclonal anti-Isl1 | DSHB | Cat# 39.3F7, RRID:AB_1157901 | IHC (1:100) |
| Antibody | Mouse monoclonal anti-Lhx1/5 | DSHB | Cat# 4F2, RRID:AB_531784 | IHC (1:1000) |
| Antibody | Guinea pig anti-Lbx1 | T. Müller (MDC, Berlin) | N/A | IHC (1:10000) |
| Antibody | Guinea pig anti-Tlx3 | T. Müller (MDC, Berlin) | N/A | IHC (1:10000) |
| Antibody | Guinea pig anti-Foxd3 | T. Müller (MDC, Berlin) | N/A | IHC (1:5000) |
| Antibody | Rabbit polyclonal anti-Pax2 | BioLegend/Covance | Cat# 901002, RRID:AB_2734656 | IHC (1:1000) |
| Antibody | Rabbit polyclonal anti-Pax2 | Thermo Fisher Scientific | Cat# 71–6000, RRID:AB_2533990 | IHC (1:1000) |
| Antibody | Sheep polyclonal anti-Chx10 | Exalpha Biologicals | Cat# X1179P RRID:AB_2567565 | IHC (1:100) |
| Antibody | Rat monoclonal anti-Ctip2 | Abcam | Cat# ab18465, RRID:AB_2064130 | IHC (1:500) |
| Antibody | Goat anti-Pax2 | R and D Systems | Cat# AF3364, RRID:AB_10889828 | IHC (1:500) |

*Continued on next page*

*Continued*

| Reagent type (species) or resource | Designation | Source or reference | Identifiers | Additional information |
|---|---|---|---|---|
| Antibody | Rabbit polyclonal anti-Satb1 | Novus | Cat# NBP2-15108 | IHC (1:1000) |
| Cell line (human) | HEK293T | Sigma-Aldrich | Cat# 12022001 | |
| Genetic reagent (*M. musculus*) | *Hb9::GFP* | The Jackson Laboratory | Cat# JAX: 005029, RRID:IMSR_JAX: 005029 | |
| Genetic reagent (*M. musculus*) | *Mir34a$^{fl/fl}$* | The Jackson Laboratory | Cat# JAX:018545, RRID:IMSR_JAX:018545 | |
| Genetic reagent (*M. musculus*) | *Mir34bc$^{-/-}$* | The Jackson Laboratory | Cat# JAX:018546, RRID:IMSR_JAX:018546 | |
| Genetic reagent (*M. musculus*) | *Mir34a$^{-/-}$* | This paper | N/A | |
| Genetic reagent (*M. musculus*) | *Mir449$^{-/-}$* | This paper | N/A | |
| Genetic reagent (*M. musculus*) | *Mir34a$^{-/-}$; Mir34bc$^{-/-}$* | This paper | N/A | |
| Genetic reagent (*M. musculus*) | *Mir34a$^{-/-}$; Mir449$^{-/-}$* | This paper | N/A | |
| Genetic reagent (*M. musculus*) | *Mir34bc$^{-/-}$; Mir449$^{-/-}$* | This paper | N/A | |
| Genetic reagent (*M. musculus*) | *Mir34a$^{-/-}$; Mir34bc$^{-/-}$; Mir449$^{-/-}$* | This paper | N/A | |
| Genetic reagent (*M. musculus*) | E2a::Cre | The Jackson Laboratory | Cat# JAX:003724, RRID:IMSR_JAX:003724 | |
| Recombinant DNA | psiCHECK-2 vector | Promega | Cat# C8021, RRID:Addgene_106979 | |
| Recombinant DNA | psiCHECK-2-Satb2-3'UTR_WT | This paper | N/A | |
| Recombinant DNA | psiCHECK-2-Satb2-3'UTR_Mut | This paper | N/A | |
| Recombinant DNA | pENTR/D-TOPO-*Mir34a* | This paper | N/A | |
| Recombinant DNA | pENTR/D-TOPO-*Mir34bc* | This paper | N/A | |
| Recombinant DNA | pENTR/D-TOPO-*Mir449abc* | This paper | N/A | |
| Recombinant DNA | p2Lox-*Mir34a* | This paper | N/A | |
| Recombinant DNA | p2Lox-*Mir34bc* | This paper | N/A | |
| Recombinant DNA | p2Lox-*Mir449abc* | This paper | N/A | |
| Software, algorithm | Adobe Illustrator 2020 | Adobe | https://www.adobe.com | |
| Software, algorithm | Adobe Photoshop 2020 | Adobe | https://www.adobe.com | |
| Software, algorithm | Image J | NIH | https://imagej.net/ RRID:SCR_003070 | |
| Software, algorithm | Imaris 9.5.1 | Bitplane | http://www.bitplane.com/imaris/imaris RRID:SCR_007370 | |
| Software, algorithm | ZEN | Carl Zeiss | RRID:SCR_013672 | |

*Continued on next page*

*Continued*

| Reagent type (species) or resource | Designation | Source or reference | Identifiers | Additional information |
|---|---|---|---|---|
| Software, algorithm | Prism 6 | GraphPad | https://www.graphpad.com RRID:SCR_005375 | |
| Software, algorithm | TreadScan4.0 | CleverSystems | N/A | |
| Software, algorithm | R and RStudio | The R foundation | https://www.r-project.org/ https://www.rstudio.com RRID:SCR_000432 | |

## Generation of Mir449 KO mice

To generate the *Mir449*$^{-/-}$ mice, we applied the CRISPR-Cas9 system using a published protocol (*Li et al., 2017*). We designed two single-guide RNAs (sgRNAs) targeting to the *Mir449* locus, as depicted in *Figure 1—figure supplement 1*. The sequences of two designed sgRNAs (*Supplementary file 1a*), together with the T7 promoter and trans-activating CRISPR RNA (tracrRNA), were concatenated during plasmid synthesis. The two *Mir449* sgRNAs and the *Cas9* mRNA were microinjected simultaneously into fertilized embryos of C57BL/6J mice. *Mir449*$^{+/-}$ mice were obtained and genotyped, with the deleted sites verified by Sanger sequencing, followed by intercross mating to acquire *Mir449*$^{-/-}$ mice (*Figure 1—figure supplement 1D-F*).

## Mouse breeding and maintenance

*Mir34a*$^{fl/fl}$ (Mir34$^{atm1.2Aven}$/J, Stock #018545), E2a::Cre (B6.FVB-Tg(EIIa-cre)C5379Lmgd/J, Stock #003724), and *Mir34bc*$^{-/-}$ (B6.Cg-Mirc21$^{tm1.1Aven}$/J, Stock #018546) mice were imported from Jackson Laboratory. *Mir34a*$^{-/-}$ mice were generated by crossing *Mir34a*$^{fl/fl}$ mice with E2a::Cre, resulting in germ-line deletion of loxP-flanked *Mir34a* followed by depletion of E2a::Cre. All individual KO mice were further crossed with each other to obtain *Mir34bc/449* DKO and *Mir34/449* TKO mice. The mutant mouse lines were maintained and bred by the following intercross matings: *Mir34a*$^{-/-}$; *Mir34bc*$^{-/-}$;*Mir449*$^{+/-}$, *Mir34a*$^{-/-}$;*Mir34bc*$^{+/-}$;*Mir449*$^{-/-}$, *Mir34bc*$^{-/-}$;*Mir449*$^{+/-}$, and *Mir34bc*$^{+/-}$;*Mir449*$^{-/-}$, as shown in *Figure 2A*. All age-matched littermates from the above matings served as a control (Ctrl) group for all experiments, unless otherwise specified. To analyze survival and body weight of newborn *Mir34bc/449* DKO, *Mir34/449* TKO, and Ctrl mice, we monitored the health of these mice every day and determined their body weight every two days for the first 21 days. The primers used for all genotyping are listed in *Supplementary file 1b*. All mice generated and used for this study had a C57BL/6J background. The smallest sample size that would still give a significant difference was employed, in accordance with 3Rs principles. No animals were involved in previous unrelated experimental procedures. The influence of mouse sex was not evaluated in this study. All the live animals were maintained in a specific-pathogen-free (SPF) animal facility, approved and overseen by IACUC Academia Sinica.

## Tissue collection

Mice were sacrificed under deep anesthesia by 20 mg/mL Avertin (2,2,2-Tribromoethanol, Sigma) with a dosage based on mouse body weight. Then, cardiac perfusion of cold PBS was performed before collecting tissues and placing in Trizol (Thermo Scientific) for RNA extraction. A subsequent perfusion was performed with freshly prepared 4% paraformaldehyde (PFA) in PBS, followed by whole spinal cord dissection, for immunostaining or *in situ* hybridization. Spinal cords were sucrose-cryoprotected and embedded in FSC 22 frozen section media (Leica), before being cut into 25 μm cryostat sections as previously described (*Tung et al., 2019*).

For embryo analyses, pregnant mice underwent cervical dislocation, and the embryos were dissected from the sacs. After removing the head and internal organs, the embryos were immersed in 4% PFA for 1 ~ 2 hr at 4°C, followed by a PBS wash. The same procedures described above for cryosectioning were used for the embryo samples, except that 20 μm cryostat sections were collected.

## microRNA *in situ* hybridization (ISH) and miRNAscope assay

For all experimental procedures, we used diethylpyrocarbonate (DEPC)-treated water or PBS for washing steps or reagent preparation. Spinal cord cryosections were initially treated with 10 µg/mL proteinase K (Invitrogen), followed by acetylation in acetic anhydride/triethanolamine, and then fixed again with 4% PFA. Next, sections were pre-hybridized in hybridization solution [50% formamide, 5 X SSC, 0.5 mg/mL yeast tRNA (Ambion), 5 X Denhardt's solution (Fisher), 0.5 mg/ml salmon sperm DNA (Thermo Fisher Scientific), 0.02% Roche blocking reagent] at room temperature for 2 ~ 4 hr, followed by hybridization with each miRNA probe overnight at 55°C. After post-hybridization washes in 2 X SSC and then 0.2 X SSC at 55°C, the *in situ* hybridization signals were detected using the NBT/BCIP (Roche) system according to the manufacturer's instructions. After termination of color development, slides were subjected to immunostaining as described below. Slides were mounted in Aqua-Poly/Mount and analyzed with a Zeiss LSM 780 confocal microscope. The 5' FITC-labeled LNA MiR34a-5p probe (ACAACCAGCTAAGACACTGCCA) was purchased from Exiqon.

To detect MiR34c-5p, spinal cord samples were dissected and fixed with 4% PFA, as described previously (*Tung et al., 2019*; *Yen et al., 2018*). The cryostat sections were processed using miRNAscope technology (Advanced Cell Diagnostics, ACD) according to the manufacturer's instructions. The probe to detect mmu-MiR34c-5p was customized from 896831-S1 (ACD).

## RNA isolation and quantitative real-time PCR (RT-qPCR)

Total RNA was isolated using the Quick-RNA MiniPrep kit (Zymo Research). For mRNA analysis, 50–300 ng of total RNA from each sample was reverse-transcribed with Superscript III (Thermo Scientific). One-tenth of the reverse transcription reaction was used for subsequent quantitative real-time PCRs on a LightCycler480 Real-Time PCR instrument (Roche) using SYBR Green PCR mix (Roche) for each gene of interest. *GAPDH* served as an internal reference for normalization of amplification efficiency. PCR primers were designed using the online database Primer3 and subjected to Basic Local Alignment Search Tool (BLAST; NCBI) analysis to avoid annealing to non-specific sequences during amplification. The sequences of the primers used are listed in *Supplementary file 1c*.

For miRNA expression analyses, 100–200 ng of total RNA from each sample was reverse transcribed with a miRNA-specific primer from TaqMan MicroRNA Assays (Life Technology). The following assays were used: MiR34a-5p (Assay ID: 000426), MiR34b-5p (Assay ID: 002617), MiR34c-5p (Assay ID: 000428), MiR449a-5p (Assay ID: 001030), MiR449c-5p (Assay ID: 001667). A ubiquitous MiR16 (Assay ID: 000391) was used as the endogenous control for normalization. Each quantitative real-time PCR was performed in duplicate per sample, with at least three different experimental samples.

## Immunostaining

Immunostaining was performed on cryostat sections as previously described (*Tung et al., 2019*; *Yen et al., 2018*). Commercially available primary antibodies used in this study are listed in Key Resources Table. Alexa488-, Cy3- and Cy5-conjugated secondary antibodies were obtained from either Invitrogen or Jackson Immunoresearch and used at 1:1000 dilutions. Images were acquired by using a Zeiss LSM710 or LSM780 confocal microscope.

## Behavioral assays

Locomotor activity was measured by the open field, rotarod, and treadmill tests, whereas nociceptive sensory responses were evaluated by the mechanically stimulated von Frey test and heat-stimulated tail flick and hot-plate tests. For the *Mir34bc/449* DKO and *Mir34/449* TKO mice, we selected surviving mutant mice as well as littermate Ctrl for the behavioral analyses. Age-matched WT mice from the same C57BL/6J background were also used for experimental comparisons. Both sexes of adult mice (~4 months-old) were used in this study. The experimenters conducting all behavioral assays were blind to mouse genotypes.

### Open-field test

A square arena with opaque walls (area 48 × 48 cm and height 35 cm) was used. Mice were transferred to the testing room and habituated in the home cage 1 hr before testing, and then allowed to explore the test chamber for a further 1 hr during which all behaviors were videotaped and

tracked using a video-tracking system mounted on top of the arena (Clever System, Reston, VA). Total distance and average velocity were analyzed for each mouse (*Wang et al., 2017*).

## Rotarod

A commercially available rotarod apparatus (47600 Rota-Rod, Ugo Basile, Italy) with a rotating rod of 5 cm diameter was used. Mice were transferred to the testing room and habituated in the home cage at least 15 min before testing. In the training phase, three trials with a constant speed of 4 rpm and a 60 s cut-off time were used to ensure that all test mice could stay on the rod for a training trial before moving to the test phase. After a 30 min rest interval, the mice were evaluated during the test phase, with the rod accelerating from 4 to 40 rpm with a 300 s cut-off time, in a series of three trials. The longest falling latency as well as the rotating speed when the mouse fell off the apparatus were used to represent the motor coordination of each mouse (*Wang et al., 2017*).

## Treadmill locomotion analysis

A TreadScan apparatus (CleverSys, Reston, VA) was used to analyze gait. Mice were placed on a stationary treadmill for acclimation and trained at a speed of 10 cm/s for 5 min before testing. Three test speeds were analyzed (15, 20, and 25 cm/s) for each trial, which were recorded at 79 frames/s for 10 s using TreadScan software. For data analyses, the successful trials in which a mouse was able to maintain treadmill speed with continuous locomotion for each 10 s recording was selected and further analyzed using TreadScan software. Only gait analyses from the trials conducted at 20 cm/s are shown in the present study. The gait parameters for each limb—including stride, stance, swing, break, and propulsion time—were automatically and unbiasedly calculated, and average values were used for statistical analysis. Representative analyses from the hindlimb were shown in this paper.

The parameters for limb coordination (phase coupling) were also calculated using TreadScan software, and the average values for homologous, homolateral, and diagonal coupling for each hindlimb from all trials were statistically analyzed. The phase coupling parameter is graphically displayed as a circular plot with phase values of 0 or 1 corresponding to perfect synchronization, whereas a phase value of 0.5 represents strict alternation. The mean phase value is indicated by the direction of the vector, and vector length represents the concentration of phase values around the mean (*Crone et al., 2009*; *Drew and Doucet, 1991*; *Kjaerulff and Kiehn, 1996*). Data visualization was performed in R.

## Von frey

Mechanosensitivity was assessed according to a previously published protocol, whereby the withdrawal threshold of the hind paws was measured by a simplified up-down method (SUDO) (*Bonin et al., 2014*; *Chuang et al., 2018*). In brief, 10 von Frey filaments with different bending forces ranging from 0.008 to 2 g were used for testing. Mice were habituated on the wire mesh 1 hr before the experiment, and then the 0.16 g filament was applied as the first stimulation. When a mouse exhibited hind paw withdrawal, lifting, flinching, or licking behaviors, this was defined as a positive response. If a positive response was shown, a lighter filament was used for the next stimulation. Conversely, a heavier filament was used for subsequent stimulation if there was a negative response to the 0.16 g filament. For each test, a constant number of five stimuli on each mouse was recorded, and the paw withdrawal threshold was determined by the fifth response and the estimated sixth response. For instance, if the fifth response was positive, then the threshold was the average of the value of the fifth response and the weight of the next lightest filament.

## Tail-flick

A commercially available tail flick apparatus (37360 Tail Flick Unit, Ugo Basile, Italy) was used to measure the nociceptive threshold of mice. The heat intensity was set at 15 units with a 22 s cut-off time. Mice were transferred to the testing room and habituated in the home cage at least 15 min before the test. Then, the test mouse was placed on the apparatus and gently restrained for the experiment. The latency to observing the test mouse flicking its tail was recorded.

### Hot-plate

A commercially available hot-plate apparatus (35100 Hot/Cold Plate, Ugo Basile, Italy) was used to determine the thermal nociception of mice. Before the experiment, mice were transferred to the testing room and habituated in the home cage for at least 15 min, and then placed on the hot-plate set at a constant temperature of $55 \pm 0.2°C$. The latency to observing the mouse display hind paw licking/flicking or jumping behaviors was recorded. A 30 s cut-off time was used to avoid the test mice suffering tissue damage.

## Mir34/449 plasmid construction

Three mouse DNA fragments encompassing the *Mir34a*, *Mir34bc*, and *Mir449abc* clusters (700 bp,~1.2 Kb, and ~2 Kb in size, respectively) were amplified from mouse genomic DNA by using 2X PCR Dye Master Mix II (ADPMX02D-100) or Phusion High-Fidelity DNA Polymerase (F-530L; Thermo Fisher Scientific), and cloned into the pENTR/D-TOPO vector (K2400-20; Life Technology) according to the manufacturer's instructions. The primers used for TOPO cloning are listed in *Supplementary file 1d*. We then performed GATEWAY recombination (11791019; Thermo Fisher Scientific) to transfer *mmu-Mir34a*, *mmu-Mir34bc*, or *mmu-Mir449abc* into a Dest plasmid, p2Loxa (*Iacovino et al., 2011*), under a PGK promoter.

## Predicted MiR34/449 target genes, putative binding sites, and gene ontology analysis

To identify putative MiR34/449-regulated genes, we used the online database TargetScan (Release 7.1 or 7.2, http://www.targetscan.org/mmu_72/). To narrow down selected candidates, we conducted gene ontology analysis via Database for Annotation, Visualization, and Integrated Discovery (DAVID) (*Huang et al., 2009*). Functional clustering annotated from DAVID, expressed as gene enrichment, is presented in this study (*Tung et al., 2015*).

## Dual luciferase reporter assay

The WT 3'UTR sequence containing four putative MiR34/449-binding sites of the *Satb2* gene was amplified from mouse genomic DNA by PCR using 2X PCR Dye Master Mix II (ADPMX02D-100) or Phusion High-Fidelity DNA Polymerase (F-530L; Thermo Fisher Scientific). The purified PCR products were then inserted into the psiCHECK-2 vector (C8021; Promega) at XhoI and NotI restriction sites by using T4 DNA ligase (M0202S; NEB). The mutated versions (Mut) of the *Satb2* 3'UTR were cloned by designing primers that resulted in the complementary sequence of the predicted binding site mismatching with the MiR34/449 seed sequence. Each fragment possessing a mutated binding site was initially amplified by PCR, followed by overlapping extension PCR, and the four mutated binding sites within the *Satb2* 3'UTR were then cloned into the psiCHECK-2 vector described above. The primers used for the WT and Mut *Satb2* 3'UTR reporter are listed in *Supplementary file 1e*.

HEK293T cells were plated at a density of $5 \times 10^4$ per well (24-well plate), expanded for 16 ~ 20 hr, and co-transfected with a mixture of 60 ng of WT or Mut reporter and 2 µg of either *Mir34a*, *Mir34bc*, *Mir449abc*, or control plasmids using 2 µL of PLUS Reagent and 2 µL of Lipofectamine LTX Reagent (A12621; Invitrogen). After 24 hr, cells were lysed and processed for luciferase assay using the Dual-Luciferase Reporter Assay System (E1910; Promega) according to the manufacturer's instructions. We measured luciferase activity using a 20/20 n luminometer (Turner Biosystems). To normalize transfection efficiency, luciferase activity was calculated as the ratio of firefly to Renilla luciferase activity, and the relative luciferase activity was further expressed as the ratio of measured luciferase activity to the control (*Chen et al., 2011*; *Tung et al., 2015*).

## HEK293T cell culture

HEK293T cells were cultured in DMEM (Gibco) supplemented with 2 mM L-Glutamine (Invitrogen), 1% Penicillin/Streptomycin (Invitrogen), and 10% FBS (Gibco) at 37°C in a humidified incubator with 5% $CO_2$. All cell lines used in this study are subjected to regular mycoplasma tests.

## Spatial analysis of spinal interneurons

The positioning of Satb2[on] INs in cervical/brachial spinal segments of P14 mice was analyzed in Imaris 9.5.1 (Bitplane) using the 'Spots' function. Cartesian coordinates were constructed for each

image, with the midpoint of the central canal defined as position (0,0). The position of each individual neuron was exported from Imaris as a. csv file and visualized using a custom R script (http://www.r-project.org), with the dorso-ventral and medio-lateral density distributions shown alongside the respective figures. To estimate the probability density of Satb2$^{on}$ INs in the transverse spinal cord, we computed two-dimensional kernel density estimations using the 'kde2d' function from the 'MASS' library in R. Estimates were displayed as contour plots of density values in the 30th–90th percentiles (at intervals of 10%), as described previously (*Bikoff et al., 2016*; *Paixão et al., 2019*; *Sweeney et al., 2018*; *Tripodi et al., 2011*).

## Statistical analysis

Results are expressed as mean ± SD (standard deviation). Each experiment was repeated at least three times to confirm the findings unless otherwise specified. Statistical analysis was performed in Graph Pad Prism 6.0 (GraphPad Software). Multiple groups were analyzed by one-way or two-way analysis of variance (ANOVA) with Tukey's multiple comparisons test as indicated in the figure legends. Two groups were analyzed by Student's t-test. p-Values of less than 0.05 were considered significant. Kaplan-Meier curves were used to analyze mouse survival, with statistically significant differences assessed by the log-rank test.

## Acknowledgements

We thank the Transgenic Core Facility in IMB (AS-CFII-108–104), Academia Sinica for invaluable technical advice and help in generating *Mir449* KO mice. Particular thanks are due to the Taiwan Mouse Clinic (MOST 107–2319-B-001–002) for their technical support in the open field, rotarod, von Frey, tail flick, and hot-plate experiments. The Lbx1, Tlx3, and Foxd3 antibodies were generous gifts from Thomas Müller and Carmen Birchmeier. We appreciate the Genomic, FACS, Bioinformatics and Imaging core facilities of IMB for considerable technical assistance. We also acknowledge members of the JAC lab for proofreading, Ya-Yin Tsai and Yi-Han Lee for cloning luciferase constructs, and J O'Brien for further reviewing the manuscript. This work is funded by Academia Sinica (CDA-107-L05 and AS-GC-109–03), MOST (109–2314-B-001–010-MY3, 109–2326-B-001–017-, 108–2311-B-001–011-) and NHRI (NHRI-EX110-10831NI).

## Additional information

### Funding

| Funder | Grant reference number | Author |
| --- | --- | --- |
| Ministry of Science and Technology, Taiwan | 110-2811-B-001-508 - | Jun-An Chen |
| Ministry of Science and Technology, Taiwan | 109–2314-B-001–010-MY3 | Jun-An Chen |
| Ministry of Science and Technology, Taiwan | 109–2326-B-001–017 | Jun-An Chen |
| Ministry of Science and Technology, Taiwan | 108-2311-B-001-011- | Jun-An Chen |
| Ministry of Science and Technology, Taiwan | 107-2311-B-001-043- | Jun-An Chen |
| Academia Sinica | CDA-107-L05 | Jun-An Chen |
| Academia Sinica | AS-GC-109–03 | Jun-An Chen |
| National Health Research Institutes | NHRI-EX110-10831NI | Jun-An Chen |

The funders had no role in study design, data collection and interpretation, or the decision to submit the work for publication.

## Author contributions
Shih-Hsin Chang, Conceptualization, Data curation, Software, Formal analysis, Validation, Investigation, Visualization, Methodology, Writing - original draft, Writing - review and editing; Yi-Ching Su, Data curation, Validation, Methodology; Mien Chang, Resources, Validation, Methodology; Jun-An Chen, Conceptualization, Data curation, Supervision, Funding acquisition, Project administration, Writing - review and editing

## Author ORCIDs
Shih-Hsin Chang ![ORCID] https://orcid.org/0000-0003-2821-7095
Yi-Ching Su ![ORCID] http://orcid.org/0000-0002-2760-9540
Jun-An Chen ![ORCID] https://orcid.org/0000-0001-9870-3203

## Decision letter and Author response
Decision letter https://doi.org/10.7554/eLife.63768.sa1
Author response https://doi.org/10.7554/eLife.63768.sa2

## Additional files

### Supplementary files
• Supplementary file 1. Primers, genotypes of mouse littermate controls, and gene list used in this study are provided in the tables.

• Transparent reporting form

### Data availability
All data generated or analysed during this study are included in the manuscript and supporting files. Source data files have been provided for supplementary video1 and 2. The R analysis script written for this paper is available at https://gitlab.com/jaclab/mir-34_449 (copy archived at https://archive.softwareheritage.org/swh:1:rev:716ab1286665e8240249bc153ee766fa5e3fa94f).

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
