## [Decision Letter]

**Acceptance summary:**

Your paper provides a very nice addition to the literature on how miRNA affects neuronal patterning and, eventually, behavior.

**Decision letter after peer review:**

Thank you for submitting your article "MicroRNAs Mediate Precise Spinal Interneuron Population to Exert Delicate Sensory-to-Motor Outputs" for consideration by *eLife*. Your article has been reviewed by two peer reviewers, and the evaluation has been overseen by a Reviewing Editor and Marianne Bronner as the Senior Editor. The reviewers have opted to remain anonymous.

The reviewers have discussed the reviews with one another and the Reviewing Editor. We agree that your manuscript is of potential interest for *eLife*, but as described below, additional experiments are required before it is published.

In brief, while we agree that additional genetic mouse manipulations and behavioral assays (such as satb2 overexpression to test for behavioral phenotypes or a genetic epistasis experiments) are not required, we believe that it is absolutely essential to perform a more thorough analysis of the ectopic satb2 positive population. You will see additional comments that you may find helpful in further improving the manuscript.

Reviewer #1:

The current manuscript by Chang et al. investigated the effect of knocking out miR-34/449 on the behavioral output from the spinal cord. After validating miR-34/449 expression in the spinal cord, the authors did a thorough job analyzing motor neuron (MN) development and function in miR-34/449 KO mice. While the results indicated that miR-34/449 KO did not lead to any significant defects in MN, the authors did notice certain behavioral changes (hyper-jumpy and hypersensitivity to thermal stimulation) that may be due to changes in interneuron (IN) function. The authors then identified an increased population of Satb2b (a target of miR-34/449)-positive interneurons in mice. The aspect of miRNA depletion and the associated behavioral phenotype is interesting. The manuscript in its current form, however, will significantly benefit from clarifications on the link between the molecular phenotype (Satb2) and behavioral phenotype.

1. Among the battery of all the behavioral tests, the hyper-jumpy and hypersensitivity to thermal stimulation appear to be the only phenotypes detected. However, since this phenotype's etiology can come from many sources, it is difficult to discern the observed behavioral output's specificity. As this is the manuscript's primary gist, the authors should further address the specificity of miRNA-target interruption and behavioral output. For example, regarding the increased Satb2-positive cells, is this finding a reflection of increased IN number versus ectopic expression of Satb2 in other cell types due to miR-34/449 being absent? It would be informative to define the nature of these increased Sabt2-positive population. Since there is insufficient evidence supporting the mechanistic link between Satb2 upregulation and the behavioral phenotype, the authors should test if the behavioral change can be mimicked by directly upregulating Satb2 in INs. Or another way of addressing this question would be to see whether reducing Satb2 in miR-34/449 KO would rescue the behavioral phenotype. Without additional experiments similarly designed to lay out the molecular connection between Satb2 dysregulation and behavioral changes, the observations stated in the manuscript may stay inconclusive.

2. The rationale for generating compete miR-34/449 KO mice (page 10) appears to be weak as the data showing the compensatory expression of miR-449 and miR-34bc shown in Supplementary Figure 1 fall short of convincing because of the large error bars, data variabilities, and the lack of statistical significance.

3. It seems that deleting miR-449 in the intronic regions affected Cdc20b expression levels, as shown in the Supplementary data. It is surprising that while miR-449 appears to be enriched in the nervous system, the deletion does not influence Cdc20b in the nervous system. Accordingly, there are some questions associated with this phenomenon. Is there any evidence that the nervous system does not use Cdc20b as a host gene for miR-449, unlike other tissues? Also, what is the expression pattern level of Cdc20b in other tissues as tested in Figure 1E and F? It is important to clarify this point to rule out the possibility of ectopic Cdc20b having a role in the spinal cord.

Reviewer #2:

The manuscript examines the role of miR-34/449 family in spinal cord development and motor circuit formation. The authors demonstrate that miR-34 family of microRNAs increased their expression in postnatal spinal cord, while miR-449 family is highly expressed only in early embryonic stages. The authors generated TKO mice in which all members of the miR-34/449 family are knocked out. Analysis of sensorimotor behaviors revealed that despite strong expression of miR-34 family in postmitotic motor neurons, mutant mice do not exhibit significant defects in gait or in the number of spinal motor neuron subtypes. Analysis of sensorimotor pathway revealed a sensitization of a spinal reflex and reduced latency in a tail flick assay. Finally, the authors describe a significant increase in the number of Satb2 premotor dorsal spinal interneurons in DKO and TKO mice and demonstrated that Satb2 is a direct target of the miR-34/449 family.

While Satb2 premotor interneurons that receive primary or secondary sensory inputs are well positioned to play a role in sensitization of the reflex circuit, in the absence of conditional deletion of miRNAs in different neuronal populations, or cell type specific rescue experiments, this conclusion remains only correlative. Nevertheless, genetic dissection of the pathway seems beyond the scope of the current manuscript.

1. Quantification of individual miRNAs by qPCR needs to be normalized to a ubiquitous miRNA that can serve as a loading control. The relative quantification does not provide satisfactory insight in the absolute expression levels of individual miRNAs. The authors provide convincing images of miR-34a expression in postmitotic neurons, but no other members of the family are examined with cellular resolution. In order to interpret contribution of this family to Satb2 regulation, it would be informative to examine expression of individual miRNA members in Satb2 interneurons and their progenitors.

2. Satb2 interneurons were previously shown to be subdivided into a medial population that receives preferentially Pvalb+ proprioceptive sensory inputs and a lateral population receiving nociceptive inputs (Hilde et al., 2016). The medial population also co-expresses CTIP2. Do the authors observe increase in both Satb2 IN subtypes?

3. Loss of Satb2 results in a lateral shift of the interneuron cell bodies. Does the position of Satb2 neurons change in the TKO animals? Is it possible that TKO interneurons expressing presumably higher levels of Satb2 become localized more medially, resulting in a local increase in the number of Satb2 positive cells?

4. Do the supernumerary Satb2 positive cells express markers of other interneurons present in the intermediate or dorsal spinal cord (consistent with the possibility that the loss of miRNAs results in ectopic upregulation of Satb2 expression)?

5. Numbers of interneurons vary along the rostro caudal aspect of the spinal cord. Besides the included quantifications performed in cervical spinal cord, the authors should examine the number of Satb2 interneurons specifically in the spinal cord segment that is engaged in the tail flick reflex response.

6. Normal numbers of Satb2 expressing cells in embryonic spinal cord, indicate that the postnatal phenotype is likely due to the miR-34 family loss of function. Do DKO mice in which miR-34a,b,c are deleted exhibit increased Satb2 numbers and sensitized spinal reflexes?

---

## [Author Response]

Reviewer #1:The current manuscript by Chang et al. investigated the effect of knocking out miR-34/449 on the behavioral output from the spinal cord. After validating miR-34/449 expression in the spinal cord, the authors did a thorough job analyzing motor neuron (MN) development and function in miR-34/449 KO mice. While the results indicated that miR-34/449 KO did not lead to any significant defects in MN, the authors did notice certain behavioral changes (hyper-jumpy and hypersensitivity to thermal stimulation) that may be due to changes in interneuron (IN) function. The authors then identified an increased population of Satb2b (a target of miR-34/449)-positive interneurons in mice. The aspect of miRNA depletion and the associated behavioral phenotype is interesting. The manuscript in its current form, however, will significantly benefit from clarifications on the link between the molecular phenotype (Satb2) and behavioral phenotype.1. Among the battery of all the behavioral tests, the hyper-jumpy and hypersensitivity to thermal stimulation appear to be the only phenotypes detected. However, since this phenotype's etiology can come from many sources, it is difficult to discern the observed behavioral output's specificity. As this is the manuscript's primary gist, the authors should further address the specificity of miRNA-target interruption and behavioral output. For example, regarding the increased Satb2-positive cells, is this finding a reflection of increased IN number versus ectopic expression of Satb2 in other cell types due to miR-34/449 being absent? It would be informative to define the nature of these increased Sabt2-positive population. Since there is insufficient evidence supporting the mechanistic link between Satb2 upregulation and the behavioral phenotype, the authors should test if the behavioral change can be mimicked by directly upregulating Satb2 in INs. Or another way of addressing this question would be to see whether reducing Satb2 in miR-34/449 KO would rescue the behavioral phenotype. Without additional experiments similarly designed to lay out the molecular connection between Satb2 dysregulation and behavioral changes, the observations stated in the manuscript may stay inconclusive.

We thank Reviewer #1 for this insightful suggestion to elaborate on the mechanistic link between Satb2 IN upregulation and the hypersensitive spinal reflex by means of genetic evidence. Given that Reviewer #2 also suggested that we explore the increase in spinal INs of *miR-34/449* mutants in greater detail, we have systematically examined other intermediate spinal IN populations, such as Satb1, Pax2, and Ctip2, in our *miR-34/449* mutant mice (Levine et al., 2014). Interestingly, we uncovered a drastic increase in Satb1^on^ INs at postnatal stage P14 in the *miR-34/449* TKO mice, with a concomitant increase in hybrid Satb1/2^on^ INs (new revised Figure 5E and 5H). This increase in Satb1^on^ and Satb2^on^ INs appears to be specific, as numbers of other proximal spinal INs, such as Pax2^on^, remained comparable in the *miR-34/449* DKO and TKO mice (new Figure 6A and 6C). Closer examination of *in silico*-predicted miR-34/449 targets further revealed that Satb1 might be a potential spinal IN target of miR-34/449 (new Figure 5D and new Supplementary file 1g). Moreover, our luciferase assay with Satb1 seed binding-sequence mutations (new Figure 5D and Figure 5—figure supplement 1B) corroborates that *Satb1* is a direct miR-34/449 target (new revised Figure 5D). Together, these new results indicate that miR-34/449 might target multiple cardinal spinal IN regulators in the motor synergy encoder (MSE) neurons of the intermediate regions of mouse spinal cord to tune the neural pathways for voluntary and reflexive movement. We now discuss how the miR-34/449-Satb1/2 axis might be involved in tuning the sensorimotor circuit, yet caution that further genetic manipulations are necessary to verify a direct link (new page 27):

“The complex behavioral repertoire of animals is regulated by a set of motor synergy encoder (MSE) neurons, in which both Satb1 and Satb2 are prominent molecular regulators (Levine et al., 2014). […] This scenario highlights the potential involvement of miRNAs in the control of mediating intricate balance of the MSE neurons formation, as well as the critical involvement of miRNAs in maintaining spinal neural circuits at the postnatal stage (Imai et al., 2016).”

Reviewer #1 further suggested an elegant genetic approach to dissect the direct involvement of Satb2 in the hypersensitive behavior of the miR-34/449 mutants. We recognize that it is an important issue, yet we also feel it might be challenging to address this concern for several reasons. First, based on our new results mentioned above, miR-34/449 seems to have multiple spinal intermediate IN targets, including Satb1. Importantly, Satb1 is also classified as an MSE neuron (like Satb2) for which exact functions have not yet been fully deciphered (Levine et al., 2014). Consequently, knockdown of Satb2 alone in the *miR-34/449* DKO or TKO mice might not elicit an obvious rescue effect. Second, we observed an increase in ectopic Satb2^on^ INs at the postnatal stage, i.e., when the connections of the spinal circuitry have been (or are being) established. *Satb2* KO mice also display aberrant IN positions and abnormal limb hyperflexion responses to mechanical and thermal stimuli (Hilde et al., 2016). Therefore, it would be strenuous to design Satb2 KD experiments at which critical developmental point and what KD levels shall reach to restore the optimal Satb2 level in the miR-34/449 mutant mice. Excessive Satb2 knockdown results in a prominent spinal hyper-reflex phenotype (Hilde et al., 2016), raising the confounding scenario that a negative rescue result might simply be due to inappropriate Satb2 knockdown. Thus, we would not be able to completely rule out other aberrant regulatory pathways that might account for the “jumpy” and hypersensitive behaviors of the *miR-34/449* null mutants. While we fully agree that this is an important issue, Reviewer #2 has also indicated that it might be beyond the scope of the current manuscript to resolve this issue. Therefore, we have added a new discussion paragraph in the revised manuscript where we acknowledge that future studies are warranted to dissect this issue further (new page 34):

“In addition, the observed neurological defects in miR-34/449 mutant mice may also have arisen from proprioceptive impairment since previous study has revealed that Satb2^on^ INs reside in the terminal region of proprioceptive afferents (Hilde et al., 2016). […] Although we have reported an increase in the number and dense positioning of the Satb2^on^;Ctip2^on^ IN subpopulation upon miR-34/449 abrogation, further assessments are required to validate a link between the proprioceptive afferents displaying this molecular change in IN identity and the peculiar mouse behaviors we observed.”

2. The rationale for generating compete miR-34/449 KO mice (page 10) appears to be weak as the data showing the compensatory expression of miR-449 and miR-34bc shown in Supplementary Figure 1 fall short of convincing because of the large error bars, data variabilities, and the lack of statistical significance.

We appreciate this criticism. To account for the variation among spinal cord sections arising from dissection, we repeated this experiment using all samples (new revised Figure 1—figure supplement 1F~H) and applied appropriate statistical analysis in the revised manuscript. We found a significant compensatory effect among the miR-34/449 miRNA members in the miR-34a/34bc DKO and miR-34a/449 DKO spinal cords.

3. It seems that deleting miR-449 in the intronic regions affected Cdc20b expression levels, as shown in the Supplementary data. It is surprising that while miR-449 appears to be enriched in the nervous system, the deletion does not influence Cdc20b in the nervous system. Accordingly, there are some questions associated with this phenomenon. Is there any evidence that the nervous system does not use Cdc20b as a host gene for miR-449, unlike other tissues? Also, what is the expression pattern level of Cdc20b in other tissues as tested in Figure 1E and F? It is important to clarify this point to rule out the possibility of ectopic Cdc20b having a role in the spinal cord.

We thank the reviewer for bringing this issue to our attention and apologize for not being clearer in our original submission. Although miR-449 appears to be enriched in the spinal cord relative to other tissues in the central nervous system, its expression is much weaker compared to miR-34a and miR-34bc based on our previously published small RNA-seq datasets (Li et al., 2017). Furthermore, expression of miR-449 is much higher in the testis compared to that in any CNS tissue (Lize et al., 2010). We think the contribution of Cdc20b upregulation to the spinal cord defect in the *miR-34/449* mutants might be minimal, as the control groups we used in this study comprise many miR-449 KO littermates in which Cdc20b was upregulated (new revised Supplementary file 1f), yet those littermates manifested no obvious hypersensitive or “jumpy” phenotypes. Most importantly, despite strong expression of miR-449 in the testis, miR-449 KO mice did not display any gross phenotype and the mutant mice were fertile.

Reviewer #2:The manuscript examines the role of miR-34/449 family in spinal cord development and motor circuit formation. The authors demonstrate that miR-34 family of microRNAs increased their expression in postnatal spinal cord, while miR-449 family is highly expressed only in early embryonic stages. The authors generated TKO mice in which all members of the miR-34/449 family are knocked out. Analysis of sensorimotor behaviors revealed that despite strong expression of miR-34 family in postmitotic motor neurons, mutant mice do not exhibit significant defects in gait or in the number of spinal motor neuron subtypes. Analysis of sensorimotor pathway revealed a sensitization of a spinal reflex and reduced latency in a tail flick assay. Finally, the authors describe a significant increase in the number of Satb2 premotor dorsal spinal interneurons in DKO and TKO mice and demonstrated that Satb2 is a direct target of the miR-34/449 family.While Satb2 premotor interneurons that receive primary or secondary sensory inputs are well positioned to play a role in sensitization of the reflex circuit, in the absence of conditional deletion of miRNAs in different neuronal populations, or cell type specific rescue experiments, this conclusion remains only correlative. Nevertheless, genetic dissection of the pathway seems beyond the scope of the current manuscript.

We appreciate the reviewer’s points and recognition of the arduous work to further dissect the genetic pathway of the miR-34/449 TKO mice we used in this study. Despite adding several new experiments to clarify this issue in the revised manuscript, we also acknowledge that the data we present do not fully elucidate a direct link between the miR-34/449-Satb1/2 axis and the hypersensitivity phenotype. Accordingly, we have added a new paragraph in the Discussion, in which we state that future studies are warranted to fully dissect this issue (page 34):

“In addition, the observed neurological defects in miR-34/449 mutant mice may also have arisen from proprioceptive impairment since previous study has revealed that Satb2^on^ INs reside in the terminal region of proprioceptive afferents (Hilde et al., 2016). […] Although we have reported an increase in the number and dense positioning of the Satb2^on^;Ctip2^on^ IN subpopulation upon miR-34/449 abrogation, further assessments are required to validate a link between the proprioceptive afferents displaying this molecular change in IN identity and the peculiar mouse behaviors we observed.”

1. Quantification of individual miRNAs by qPCR needs to be normalized to a ubiquitous miRNA that can serve as a loading control. The relative quantification does not provide satisfactory insight in the absolute expression levels of individual miRNAs. The authors provide convincing images of miR-34a expression in postmitotic neurons, but no other members of the family are examined with cellular resolution. In order to interpret contribution of this family to Satb2 regulation, it would be informative to examine expression of individual miRNA members in Satb2 interneurons and their progenitors.

We appreciate the reviewer’s criticism. In Figure 1, the miR-34/449 miRNAs were firstly normalized against ubiquitously expressed miR-16 in the corresponding samples (delta Ct). Then, we further compared relative expression (delta-delta Ct) between different tissues (Figure 1B~F) or in the spinal cord at various stages (Figure 1G~K). To compare different tissues, we used cortex as the normalization control and the data are presented as fold changes. For developmental stage comparisons, postnatal day 1 (P1) was used as the normalization standard for generating fold change. We have now revised the figure legend to clarify how relative expression of miR-34/449 was quantified.

To examine the expression patterns of the remaining miR-34/449 members at the cellular level, we made several attempts to perform standard *in situ* hybridization on miR-34bc and miR-449a, but without success. Perhaps that outcome is not surprising given that expression of miR-34b/c and miR-449a/b/c is much weaker relative to miR-34a in spinal INs (Li et al., 2017). As an alternative, we adopted an miRNAscope assay for *in situ* hybridization of miR-34c (our probe sequence does not differentiate miR-34b/c) and miR-449a, with adjacent immunostainings with Satb1/2, at postnatal stage (P14) (new Figure 5—figure supplement 2B). Whereas there was no obvious positive signal for miR-449a at P14 (data not shown), we found that miR-34b/c signal largely overlapped with that of its target Satb1/2. In doing so, we have corroborated that miR-34a/b/c and Satb1/2 are co-expressed, meaning that the former might directly target the latter to tune optimal production of MSE neurons at the postnatal stage.

2. Satb2 interneurons were previously shown to be subdivided into a medial population that receives preferentially Pvalb+ proprioceptive sensory inputs and a lateral population receiving nociceptive inputs (Hilde et al., 2016). The medial population also co-expresses CTIP2. Do the authors observe increase in both Satb2 IN subtypes?

We very much appreciate this insightful suggestion from Reviewer #2. We have now conducted immunostaining for both Satb2 and Ctip2 in the miR-34/449 mutant spinal cord to establish if there is a change in Satb2 IN subpopulations relative to Ctrl. We found increased numbers of the Ctip2 subpopulation (Satb2^on^;Ctip2^on^) in the miR-34/449 TKO spinal cord. Moreover, when we normalized numbers across all Satb2^on^ IN population, the percentage of the Ctip2^on^ subpopulation was increased in the TKO line, suggesting that miR-34/449 abrogation preferentially increases the Ctip2^on^ subtype of Satb2^on^ INs. These new results are shown in Figure 6, together with correspondingly revised text in the Results section on page 23. We also discuss this phenotype and its implications (page 28) as follows:

“In cortical neurons, Satb2 inhibits Ctip2 activity to define two major classes of projection neurons (Alcamo et al., 2008; Britanova et al., 2008). […] In this study, we have unveiled that numbers of Satb2^on^;Ctip2^on^ spinal INs increase upon miR-34/449 KO in mice, raising the possibility that one major function of miR-34/449 is to refine the optimal MSE neurons for precise sensory-motor outputs. Given the prominent activity-dependent and tuning role of miRNAs (Chen and Chen, 2019; Sim et al., 2014), it is tantalizing to test in the future if miR-34/449 also establishes the balance of Satb2^on^;Ctip2^on^ INs among cortical projection neurons.”

3. Loss of Satb2 results in a lateral shift of the interneuron cell bodies. Does the position of Satb2 neurons change in the TKO animals? Is it possible that TKO interneurons expressing presumably higher levels of Satb2 become localized more medially, resulting in a local increase in the number of Satb2 positive cells?

We thank Reviewer #2 for this insightful suggestion to assess any positional change in the increased population of Satb2^on^ INs and to determine if supernumerary Satb2^on^ INs are affected in terms of subtypes or other INs in the miR-34/449 mutant spinal cord. Firstly, to determine the spatial distribution of the Satb2^on^ INs in the postnatal spinal cord, we identified the positioning of all Satb2^on^ cells and present the results as a contour density plot (new Figure 5—figure supplement 5A and 5B). We found that most of the Satb2^on^ cells are located in the medial region, with only a few cells extending into more lateral regions, as has been reported previously (Hilde et al., 2016; Levine et al., 2014). However, this ectopic positional change in the miR-34bc/449 DKO and miR-34/449 TKO spinal cord compared to the WT and Ctrl groups did not reach statistical significance. This result is consistent with our observation that the increased subpopulation of Satb2^on^;Ctip2^on^ INs are primarily localized in the medial part of the spinal cord. These new results are presented in Figure 5—figure supplement 5A and 5B, and the correspondingly revised text can be found in the Results section (page 23). We also explain our results further in the Discussion (page 28) as follows:

“In cortical neurons, Satb2 inhibits Ctip2 activity to define two major classes of projection neurons (Alcamo et al., 2008; Britanova et al., 2008). […] Given the prominent activity-dependent and tuning role of miRNAs (Chen and Chen, 2019; Sim et al., 2014), it is tantalizing to test in the future if miR-34/449 also establishes the balance of Satb2^on^;Ctip2^on^ INs among cortical projection neurons.”

4. Do the supernumerary Satb2 positive cells express markers of other interneurons present in the intermediate or dorsal spinal cord (consistent with the possibility that the loss of miRNAs results in ectopic upregulation of Satb2 expression)?

This is an interesting question. To address it, we performed immunostainings for Satb2 and other known IN markers, and further systematically examined other intermediate spinal IN populations such as Satb1, Pax2, and Ctip2 in the miR-34/449 mutant mice (Hilde et al., 2016; Levine et al., 2014). Interestingly, we uncovered a drastic increase in Satb1^on^ INs at postnatal stage P14 in the *miR-34/449* TKO mice, with a concomitant increase in hybrid Satb1/2^on^ INs (new revised Figure 5E and 5H). This increase in Satb1^on^ and Satb2^on^ INs appears to be specific, as numbers of other proximal spinal INs, such as Pax2^on^, remained comparable in the *miR-34/449* DKO and TKO mice (new Figure 6A and 6C). Closer examination of *in silico*-predicted miR-34/449 targets further revealed that Satb1 might be a potential spinal IN target of miR-34/449 (new Figure 5D and new Supplementary file 1g). Moreover, our luciferase assay with Satb1 seed binding-sequence mutations (new Figure 5D and Figure 5—figure supplement 1B) corroborates that *Satb1* is a direct miR-34/449 target (new revised Figure 5D). Together, these new results indicate that miR-34/449 might target multiple cardinal spinal IN regulators in the motor synergy encoder (MSE) neurons of the intermediate regions of mouse spinal cord to tune the neural pathways for voluntary and reflexive movement. We now discuss how the miR-34/449-Satb1/2 axis might be involved in tuning the sensorimotor circuit, yet caution that further genetic manipulations are necessary to verify a direct link (new page 27):

“The complex behavioral repertoire of animals is regulated by a set of motor synergy encoder (MSE) neurons, in which both Satb1 and Satb2 are prominent molecular regulators (Levine et al., 2014). In contrast to Satb2, the function of Satb1^on^ INs is not yet completely understood (Hilde et al., 2016). […] This scenario highlights the potential involvement of miRNAs in the control of mediating intricate balance of the MSE neurons formation, as well as the critical involvement of miRNAs in maintaining spinal neural circuits at the postnatal stage (Imai et al., 2016).”

5. Numbers of interneurons vary along the rostro caudal aspect of the spinal cord. Besides the included quantifications performed in cervical spinal cord, the authors should examine the number of Satb2 interneurons specifically in the spinal cord segment that is engaged in the tail flick reflex response.

We appreciate this suggestion. For the tail flick reflex response, previous findings from rats spinally transected above the lumbar region revealed no interference with this response (King et al., 1997). Therefore, we examined the three spinal segments – including cervical/brachial, thoracic, and lumbar segments – for Satb2 expression and found that all three segments presented increased numbers of Satb2^on^ INs in the miR-34/449 TKO spinal cord relative to their corresponding control segments. These new results are now shown in new Figure 5—figure supplement 4, with correspondingly revised texts in the Results section on page 22. Our finding of a uniform increase in the Satb2^on^ IN population along the rostro-caudal axis may be due to enrichment and uniform distribution of miR-34a expression in whole spinal neurons, for which there is no dominant expression in any specific segment.

6. Normal numbers of Satb2 expressing cells in embryonic spinal cord, indicate that the postnatal phenotype is likely due to the miR-34 family loss of function. Do DKO mice in which miR-34a,b,c are deleted exhibit increased Satb2 numbers and sensitized spinal reflexes?

We appreciate the reviewer raising this issue. Since expression of miR-34a and miR-34bc miRNAs progressively increased in the developing spinal cord (Figure 1G~I), deletion of miR-34a/34bc may be expected to induce changes at either the molecular or behavioral level. However, at the molecular level, we did not observe any obvious increment in the Satb2^on^ IN population in the miR-34a/34bc DKO spinal cord. At the behavioral level, we did find that some Ctrl mice, including Ctrl of the miR-34a/34bc DKO group, displayed milder and lower penetrance of the phenotypes we reported in this study (such as ptosis and jumpy behavior). Regarding the hypersensitivity response to the tail flick test, we recorded two WT mice that presented a latency of < 5 sec in response to radiant heat applied to the tail (Figure 4C), suggesting that some of our control mice also displayed the sensitized spinal reflex. However, on average, both miR-34bc/449 DKO and miR-34/449 TKO mouse groups presented a more robust hypersensitivity response than the WT and Ctrl groups. We now explain these findings in our Discussion (page 31) as follows:

“Why is it that miR-34bc/449 DKO mice displayed a seemingly similar defective phenotype to miR-34/449 TKO mice, whereas the miR-34a/449 DKO mice are largely normal? Given that miR-34a seems to be expressed at high levels ubiquitously in almost all cell types, including spinal neurons (Concepcion et al., 2012; Otto et al., 2017; Song et al., 2014; Wu et al., 2014), this is a puzzling scenario. It is generally accepted that pre-microRNA is processed through Dicer to generate a miRNA duplex, consisting of miRNA and miRNA* strands (also termed guide and passenger strands, respectively). […] Interestingly, in the current study, we found that miR-34b-3p (mir-34b*) is significantly expressed in the developing spinal cord (data not shown), raising the possibility that miR-34b-3p might also be functionally relevant, even though its seed sequence is not conserved when compared to that of miR-34a-3p (mir-34a*).”

References:

Alcamo, E.A., Chirivella, L., Dautzenberg, M., Dobreva, G., Farinas, I., Grosschedl, R., and McConnell, S.K. (2008). Satb2 regulates callosal projection neuron identity in the developing cerebral cortex. Neuron *57*, 364-377.Britanova, O., de Juan Romero, C., Cheung, A., Kwan, K.Y., Schwark, M., Gyorgy, A., Vogel, T., Akopov, S., Mitkovski, M., Agoston, D., et al. (2008). Satb2 is a postmitotic determinant for upper-layer neuron specification in the neocortex. Neuron *57*, 378-392.Chen, J.A., Huang, Y.P., Mazzoni, E.O., Tan, G.C., Zavadil, J., and Wichterle, H. (2011). Mir-17-3p controls spinal neural progenitor patterning by regulating Olig2/Irx3 cross-repressive loop. Neuron *69*, 721-735.Chen, T.H., and Chen, J.A. (2019). Multifaceted roles of microRNAs: From motor neuron generation in embryos to degeneration in spinal muscular atrophy. *eLife 8*.Concepcion, C.P., Han, Y.C., Mu, P., Bonetti, C., Yao, E., D'Andrea, A., Vidigal, J.A., Maughan, W.P., Ogrodowski, P., and Ventura, A. (2012). Intact p53-dependent responses in miR-34-deficient mice. PLoS Genet *8*, e1002797.Harb, K., Magrinelli, E., Nicolas, C.S., Lukianets, N., Frangeul, L., Pietri, M., Sun, T., Sandoz, G., Grammont, F., Jabaudon, D., et al. (2016). Area-specific development of distinct projection neuron subclasses is regulated by postnatal epigenetic modifications. *eLife 5*, e09531.Hilde, K.L., Levine, A.J., Hinckley, C.A., Hayashi, M., Montgomery, J.M., Gullo, M., Driscoll, S.P., Grosschedl, R., Kohwi, Y., Kohwi-Shigematsu, T., et al. (2016). Satb2 Is Required for the Development of a Spinal Exteroceptive Microcircuit that Modulates Limb Position. Neuron *91*, 763-776.Huang da, W., Sherman, B.T., and Lempicki, R.A. (2009). Systematic and integrative analysis of large gene lists using DAVID bioinformatics resources. Nat Protoc *4*, 44-57.Imai, F., Chen, X., Weirauch, M.T., and Yoshida, Y. (2016). Requirement for Dicer in Maintenance of Monosynaptic Sensory-Motor Circuits in the Spinal Cord. Cell Rep *17*, 2163-2172.King, T.E., Joynes, R.L., and Grau, J.W. (1997). Tail-flick test: II. The role of supraspinal systems and avoidance learning. Behav Neurosci *111*, 754-767.Lai, H.C., Seal, R.P., and Johnson, J.E. (2016). Making sense out of spinal cord somatosensory development. Development *143*, 3434-3448.Levine, A.J., Hinckley, C.A., Hilde, K.L., Driscoll, S.P., Poon, T.H., Montgomery, J.M., and Pfaff, S.L. (2014). Identification of a cellular node for motor control pathways. Nature neuroscience *17*, 586-593.Li, C.J., Hong, T., Tung, Y.T., Yen, Y.P., Hsu, H.C., Lu, Y.L., Chang, M., Nie, Q., and Chen, J.A. (2017). MicroRNA filters Hox temporal transcription noise to confer boundary formation in the spinal cord. Nat Commun *8*, 14685.Lize, M., Pilarski, S., and Dobbelstein, M. (2010). E2F1-inducible microRNA 449a/b suppresses cell proliferation and promotes apoptosis. Cell death and differentiation *17*, 452-458.Okamura, K., Phillips, M.D., Tyler, D.M., Duan, H., Chou, Y.T., and Lai, E.C. (2008). The regulatory activity of microRNA* species has substantial influence on microRNA and 3' UTR evolution. Nat Struct Mol Biol *15*, 354-363.Osseward, P.J., 2nd, and Pfaff, S.L. (2019). Cell type and circuit modules in the spinal cord. Curr Opin Neurobiol *56*, 175-184.Otto, T., Candido, S.V., Pilarz, M.S., Sicinska, E., Bronson, R.T., Bowden, M., Lachowicz, I.A., Mulry, K., Fassl, A., Han, R.C., et al. (2017). Cell cycle-targeting microRNAs promote differentiation by enforcing cell-cycle exit. Proceedings of the National Academy of Sciences of the United States of America *114*, 10660-10665.Sim, S.E., Bakes, J., and Kaang, B.K. (2014). Neuronal activity-dependent regulation of MicroRNAs. Mol Cells *37*, 511-517.Song, R., Walentek, P., Sponer, N., Klimke, A., Lee, J.S., Dixon, G., Harland, R., Wan, Y., Lishko, P., Lize, M., et al. (2014). miR-34/449 miRNAs are required for motile ciliogenesis by repressing cp110. Nature *510*, 115-120.Tung, Y.T., Lu, Y.L., Peng, K.C., Yen, Y.P., Chang, M., Li, J., Jung, H., Thams, S., Huang, Y.P., Hung, J.H., et al. (2015). Mir-17 approximately 92 Governs Motor Neuron Subtype Survival by Mediating Nuclear PTEN. Cell Rep *11*, 1305-1318.Tung, Y.T., Peng, K.C., Chen, Y.C., Yen, Y.P., Chang, M., Thams, S., and Chen, J.A. (2019). Mir-17 approximately 92 Confers Motor Neuron Subtype Differential Resistance to ALS-Associated Degeneration. Cell Stem Cell *25*, 193-209 e197.Wu, J., Bao, J., Kim, M., Yuan, S., Tang, C., Zheng, H., Mastick, G.S., Xu, C., and Yan, W. (2014). Two miRNA clusters, miR-34b/c and miR-449, are essential for normal brain development, motile ciliogenesis, and spermatogenesis. Proceedings of the National Academy of Sciences of the United States of America *111*, E2851-2857.